# Enhanced silica export in a future ocean triggers global diatom decline

Jan Taucher[1 ✉], Lennart T. Bach[2], A. E. Friederike Prowe[1], Tim Boxhammer[1], Karin Kvale[1,3] & Ulf Riebesell[1]

Diatoms account for up to 40% of marine primary production[1,2] and require silicic acid to grow and build their opal shell[3]. On the physiological and ecological level, diatoms are thought to be resistant to, or even benefit from, ocean acidification[4–6]. Yet, global-scale responses and implications for biogeochemical cycles in the future ocean remain largely unknown. Here we conducted five in situ mesocosm experiments with natural plankton communities in different biomes and find that ocean acidification increases the elemental ratio of silicon (Si) to nitrogen (N) of sinking biogenic matter by $17 \pm 6$ per cent under $p_{CO_2}$ conditions projected for the year 2100. This shift in Si:N seems to be caused by slower chemical dissolution of silica at decreasing seawater pH. We test this finding with global sediment trap data, which confirm a widespread influence of pH on Si:N in the oceanic water column. Earth system model simulations show that a future pH-driven decrease in silica dissolution of sinking material reduces the availability of silicic acid in the surface ocean, triggering a global decline of diatoms by 13–26 per cent due to ocean acidification by the year 2200. This outcome contrasts sharply with the conclusions of previous experimental studies, thereby illustrating how our current understanding of biological impacts of ocean change can be considerably altered at the global scale through unexpected feedback mechanisms in the Earth system.

Global phytoplankton biogeography is tightly linked to the nutrient availability in the surface ocean[7,8]. Diatoms, the dominant group of silicifiers, sustain some of the most productive marine ecosystems and are a major driver of biological $CO_2$ sequestration in the oceans[1,9]. In contrast to most other phytoplankton taxa, they require silicic acid $(Si(OH)_4)$ for biomineralization of their opaline shells, called frustules. Therefore, the availability of Si compared to other major nutrients such as N determines the large-scale distribution of diatoms[10,11]. By gravitational sinking of biogenic particles (known as the biological pump)[12], Si and N are stripped out of the surface ocean and transported to deeper water layers, where remineralization of organic matter and chemical dissolution of biogenic opal convert them back to their dissolved forms. Thus, the stoichiometric Si:N ratio of particulate matter export (Si:N$_{export}$), in combination with physical transport by global ocean circulation, determines the large-scale distribution of inorganic nutrients over long timescales and the prevalence of diatoms in the world ocean[10,13,14]. However, potential future changes in global nutrient distributions of Si and N and how they may affect diatoms in the oceans are presently unknown.

The current state of ocean acidification (OA) research suggests that diatoms will be primarily affected on the physiological level (for example, by changing the availability of $CO_2$ for photosynthesis) or by altered ecological interactions within the plankton community. Numerous physiological and ecological studies have made it evident that OA effects on diatoms are very variable[4], probably owing to the

diversity in relevant traits (for example, cell size or carbon-uptake mechanism), as well as interactions with other environmental factors such as light, nutrients and temperature[6]. Overall, a growing body of evidence suggests that positive effects on diatoms are more common than negative effects, both on the species and community level[5]. Consequently, diatoms are considered to be among the 'winners' of OA.

## Impacts of OA on Si:N$_{export}$

We analysed data on the Si:N composition of vertical particle fluxes from five in situ mesocosm experiments, in which natural plankton communities (including diatoms) in different biomes were exposed to simulated OA (see Methods and Extended Data Fig. 1). This experimental approach enabled us to differentiate between OA effects on production (for example, responses of the diatom community) and on export/degradation (for example, changes in the composition of biogenic particles during sinking). To test for a systemic influence of OA on Si:N$_{export}$, we quantified OA effects using log-response ratios (lnRR) and probability densities for end-of-century $p_{CO_2}$ concentrations according to Representative Concentration Pathway (RCP) 6.0 to 8.5 scenarios of the fifth Intergovernmental Panel on Climate Change (IPCC) assessment report[15]. Our analysis reveals an overall increase of 17% (lnRR = 0.16) in the Si:N$_{export}$ ratio under OA conditions (Fig. 1). Significant effects occurred in 4 out of 5 studies, with very similar magnitudes (lnRR ranging from 0.14 in Sweden to 0.19 in Svalbard; see

[1]GEOMAR Helmholtz Centre for Ocean Research Kiel, Kiel, Germany. [2]Institute for Marine and Antarctic Studies, University of Tasmania, Hobart, Tasmania, Australia. [3]GNS Science, Lower Hutt, New Zealand. ✉e-mail: jtaucher@geomar.de

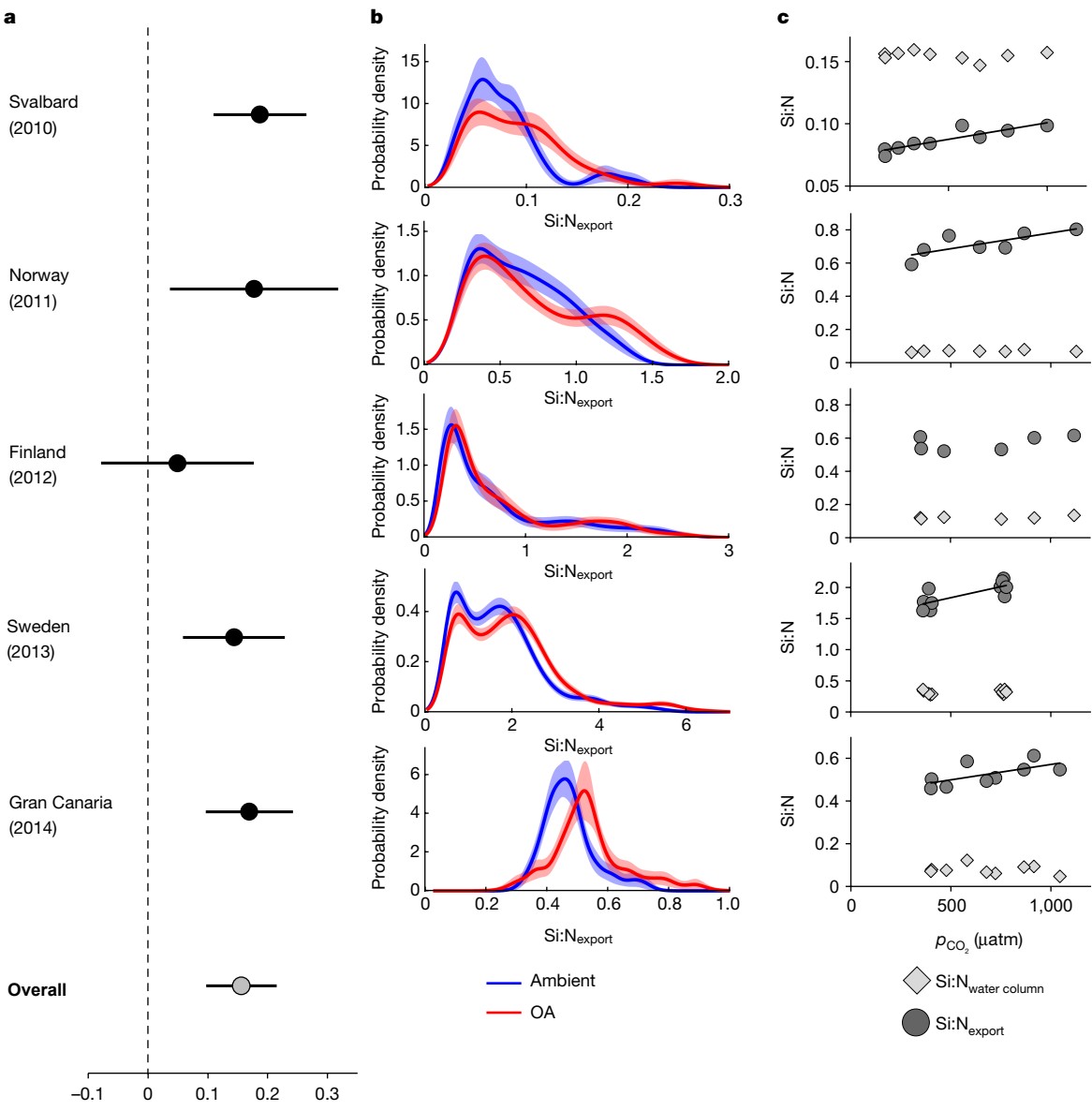

**Fig. 1 | Impacts of OA on Si:N of sinking particulate matter observed during in situ mesocosm studies in different marine biomes. a**, Effect size calculated as the log-transformed response ratio (lnRR) of the treatment averages with 95% confidence intervals, including overall effect across studies (light grey). **b**, Probability density estimates of Si:N$_{export}$ under ambient conditions (blue) and OA (red) with shaded areas denoting confidence intervals. Non-overlapping density estimates indicate statistically significant differences in the probability distributions of Si:N. Note the different scaling of

the $x$ axes, due to the variable ranges of Si:N values. **c**, Differences between OA effects on Si:N of biogenic material suspended in the water column (Si:N$_{water column}$, light grey) and sinking particulate matter collected in sediment traps (Si:N$_{export}$, dark grey). Black lines indicate that linear regression analysis was significant ($p < 0.05$). In the majority of studies (4 out of 5), we found a distinct influence of increasing $p_{CO_2}$ on the Si:N$_{export}$, but no influence on Si:N$_{water column}$. See Methods and Extended Data Table 1 for an overview of mesocosm experiments.

Methods for calculation of effect sizes). Observed shifts in Si:N$_{export}$ were surprisingly consistent across the different biomes with variable contributions of diatoms to overall plankton biomass, as indicated by the range of baseline Si:N ratios (<0.1 in Svalbard to approximately 1.8 in Sweden; Fig. 1b).

Notably, the influence of OA on Si:N was only detectable for sinking particles (collected in sediment traps), but not for particulate matter suspended in the water column (Fig. 1c). This suggests that the observed OA effects on Si:N emerged primarily while the biogenic detrital particles were sinking and not due to biotic effects during their production (see Methods). Most probably, lower pH decelerated the dissolution of sinking opal and thereby increased Si:N$_{export}$. Seawater is generally undersaturated in silicic acid and thus corrosive to

the amorphous silica of diatom shells[16]. Thus, diatoms protect their cell wall silica against chemical dissolution with an organic coating surrounding the cell[17]. Once this protective coating is degraded by bacteria (with the onset of senescence), dissolution rates of diatom silica increase drastically[18]. This usually coincides with the termination of diatom blooms and sinking of this biomass (for example, as marine snow). This mechanism explains why an OA effect on opal dissolution in our mesocosm data only becomes apparent in sinking material but not in suspended particulate matter. This interpretation is supported by both theory and empirical studies, which have demonstrated that chemical dissolution rates of biogenic silica are reduced at lower pH[17,19,20]. The observed OA effect in our experiments was a 17% increase in Si:N$_{export}$ for a pH decrease of around 0.3. Assuming that this was

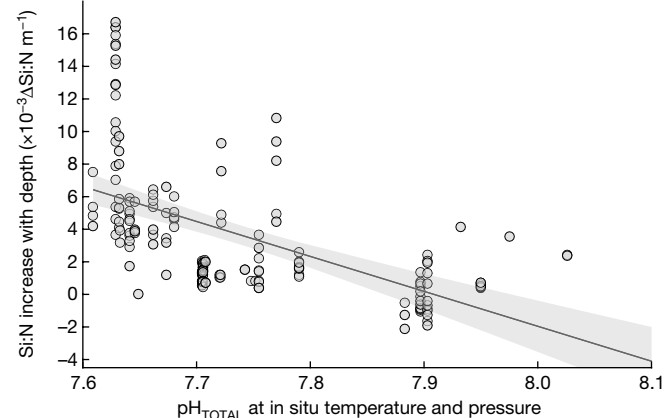

**Fig. 2 | Influence of pH on biogenic silica dissolution in the ocean.** Shown is the relationship between pH and the increase in $Si:N_{export}$ with depth ($\Delta Si:N$ m$^{-1}$) as an indicator for Si preservation and dissolution. Data are from a global compilation of sediment trap data[35]. Results from linear regression analysis reveal a significant relationship between pH and change in Si:N ratios with depth (slope = −0.0214, $R^2$ = 0.24, $p < 10^{-11}$, $f$ = 58.1, $n$ = 190). Shaded area denotes 95% confidence intervals of the regression.

solely driven by the pH sensitivity of opal dissolution, this corresponds to a decrease in opal dissolution rate of 57% per unit pH. This number agrees very well with results from chemical studies that found 60% to 70% changes in silica dissolution per unit pH[20,21]. Furthermore, these results are in line with theoretical dissolution kinetics: biogenic silica mainly consists of tetrahedrally bonded Si–O–Si silica, but also contains hydroxylated silica (for example, silanol: Si–OH). The chemical dissolution of biogenic silica is controlled by the breaking of bridging Si–O bonds, which is facilitated at higher pH owing to increased deprotonation of silanol groups at the reactive particle surface[19,22]. Although alternative explanations for the observed OA impact on Si:N cannot be fully excluded (see Methods), the various independent lines of evidence and the consistency of datasets (mesocosms, ocean sediment traps, previous chemical studies) suggest that the pH dependence of biogenic silica dissolution is the most probable explanation for our findings.

This mechanism can also explain why such an effect of OA on $Si:N_{export}$ has not been observed before, as few experimental studies have the capacity to differentiate between suspended and sinking material (that is, production and degradation of sinking particles). Furthermore, opal dissolution kinetics are probably also the reason why Finland is the only study site where no OA effect on Si:N was detected (Fig. 1): because salinity is another relevant factor controlling rates of chemical opal dissolution—higher salinity enhances dissolution rates[19,23]—the low salinity of approximately 6 psu (practical salt units) in Finland probably obscured any potential influence of OA. It should be noted that response patterns of Si:C were more variable than those detected for Si:N, owing to additional and variable shifts in C:N under OA (Extended Data Fig. 2).

To further test our interpretation that the observed OA impacts on $Si:N_{export}$ were driven by the pH sensitivity of opal dissolution, we analysed a global compilation of sediment trap data to examine the influence of pH on Si:N ratios of sinking particles in the ocean. It is widely known that opal dissolution in the water column is slower than organic matter remineralization, resulting in particulate matter that is progressively Si-enriched while sinking to depth[24,25]. Our analysis reveals that this preservation of Si compared to N (given as the increase in Si:N with depth) in the oceanic water column is enhanced at lower pH (Fig. 2). This result is consistent with our findings from our OA experiments, demonstrating that pH has a relevant role in controlling opal dissolution and the remineralization depth of silica in the ocean. Yet, this effect has so far been overlooked in the context of global-scale

OA research, because the primary factor controlling the dissolution of biogenic silica in the water column is known to be temperature[16,26] (see Methods).

## Global biogeochemical implications

The consistency of our experimental results across plankton biomes, together with the evidence for pH-dependent opal dissolution as the major driving mechanism, provide a well founded mechanistic explanation for our findings, suggesting that the sensitivity of $Si:N_{export}$ to OA is a globally relevant mechanism. We applied an Earth system model (the University of Victoria Earth System and Climate Model, UVic ESCM) to assess how changes in $Si:N_{export}$—generated by pH-sensitive opal dissolution—affect global plankton biogeography and nutrient distributions under sustained OA until the year 2200 (following extended RCP8.5 and RCP6.0 scenarios)[27]. Consistent with other Earth system models, our simulations predict profound impacts of climate change on the world ocean, for example, sea surface warming and decreased nutrient supply, which in turn lead to a decline in phytoplankton biomass (including diatoms[28,29]; see Methods). On top of that, our simulations with pH effects on opal dissolution and $Si:N_{export}$ reveal additional biogeochemical impacts of OA that entail far-reaching consequences for the ocean silica cycle and diatoms over the coming centuries. In this context, it is noteworthy that future changes in temperature and pH have antagonistic effects on opal dissolution—that is, the well known acceleration due to warming and the here-revealed slowdown in response to OA, with the effect of decreasing pH overcompensating for effects of warming (see Methods and Extended Data Fig. 3).

Slower opal dissolution under OA amplifies the silicate pump, that is, enhancing the transfer of Si from the surface to the deep ocean (Fig. 3a). This loss of $Si(OH)_4$ from the productive surface layer is strongest in the Southern Ocean, where present diatom-driven productivity and opal export are highest (Extended Data Fig. 4), and to a lesser extent in the North Pacific and Atlantic oceans. Generally, the Southern Ocean has a key role in the ocean Si cycle (i) by acting as a 'silicon trap' (about half of the global silicic acid inventory is recycled here)[14], and (ii) through the formation of mode and intermediate waters, which constitute major sources of nutrients for primary production in large parts of the global ocean via the meridional overturning circulation[10,30]. Both mechanisms are altered by OA in our simulations. Elevated $Si:N_{export}$ in effect redistributes Si to depths greater 2,000–3,000 m south of the Sub-Antarctic Front (-55° S), where it enters mainly Antarctic bottom water and partly returns towards the divergence zone of the Southern Ocean through upwelling with circumpolar deep water. This creates a cycle of particle export with elevated Si:N and slower opal dissolution and upwelling of Si-enriched circumpolar deep water in the Antarctic divergence zone, which is, however, not enough to counteract the stronger silicate pump (Fig. 3a). Together, this gradually depletes $Si(OH)_4$ in the upper 1,000–2,000 m and enhances permanent accumulation of Si ('trapping') in the deep Southern Ocean. As a consequence, Si is diminished in Antarctic mode and intermediate waters, which distribute nutrients northward within the upper limb of the global overturning circulation[10,30]. Once the Antarctic mode and intermediate waters reach the surface in the northern hemisphere (mainly in the Atlantic), these already Si-deficient water masses fuel diatom production and elevated $Si:N_{export}$ further redistributes Si from the surface to deep ocean.

The combination of these processes results in an OA-driven global loss of Si from the surface ocean and its trapping in the deep ocean. Global average $Si(OH)_4$ concentrations in the surface ocean decrease by −11% to −27% owing to OA until the year 2200 (RCP6.0 and RCP8.5, respectively). The progressive decline in $Si(OH)_4$ in the euphotic zone reduces the availability of this essential resource for diatoms, resulting in an OA-driven decline of global diatom biomass of −13% to −26% by 2200 (for RCP6.0 and RCP8.5, respectively; Fig. 3b). Almost half of

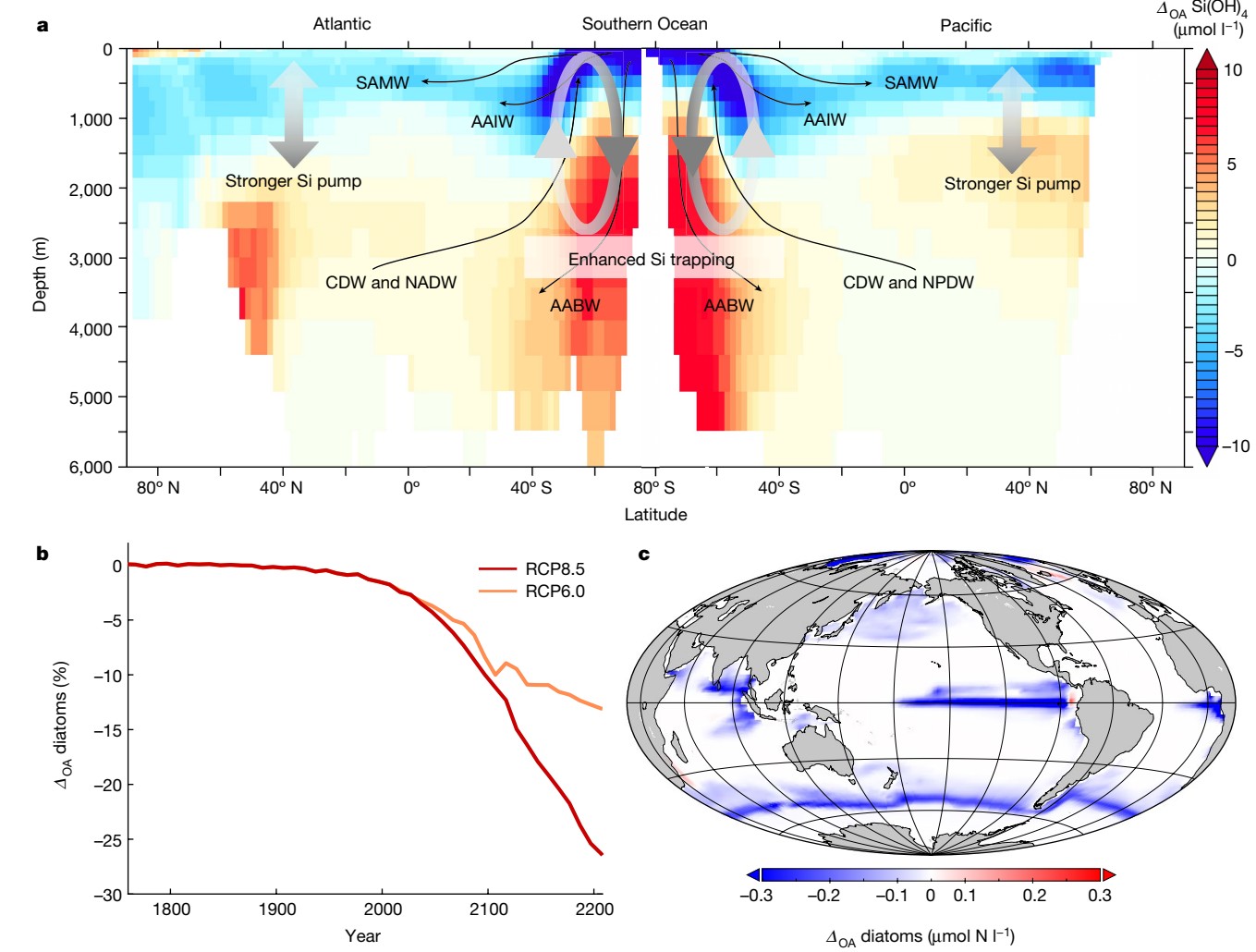

**Fig. 3 | Global impacts of slower silica dissolution under OA. a**, Differences in the global vertical distribution of $Si(OH)_4$ (zonally averaged for the Atlantic and Pacific Ocean), including a depiction of the OA-driven downward transport of Si and its trapping in the deep ocean. AABW, Antarctic bottom water; AAIW, Antarctic intermediate water; CDW, circumpolar deep water; NADW, North Atlantic deep water; NPDW, North Pacific deep water; SAMW, sub-Antarctic mode water. **b**, OA effect on global diatom biomass under RCP8.5 and RCP6.0 emission scenarios. **c**, Spatial distribution of the difference in diatom biomass. Results show the net effect of the OA-driven slowdown of silica dissolution ($\Delta_{OA}$), given as the difference to the standard model configuration (that is, including impacts of climate change, but excluding OA effects on Si dissolution). Panels **a** and **c** show results from year 2200 of the RCP8.5 simulation. See Extended Data Table 3 and Extended Data Fig. 5 for additional results and visualizations.

this decline occurs at the northern boundary of their Southern Ocean habitat around 55° S (Fig. 3c), where $Si(OH)_4$ was already limiting under preindustrial conditions (Extended Data Fig. 4) and becomes progressively depleted owing to OA-enhanced $Si:N_{export}$. We note that the above numbers refer to the net effect of OA-driven changes in silica dissolution that occur on top of other impacts of climate change—for example, reduced nutrient supply due to enhanced thermal stratification (see Methods and Extended Data Table 3 for additional results).

Although these results should be considered as a first estimate of how OA impacts on $Si:N_{export}$ may affect the future biogeography of diatoms, we emphasize that the simulated global decline of diatoms is driven by an abiotic mechanism, thereby substantially reducing the uncertainty that is usually inherent to the complex and variable biotic responses to OA[31–33]. Therefore, it is reasonable to assume that the OA impacts on $Si:N_{export}$ and opal dissolution would yield similar results if incorporated into other global models. However, associated consequences for ecosystem functioning and carbon cycling are more difficult to assess, as our simulations (similar to most Earth system models) do not account for potentially relevant physiological and ecological mechanisms that might trigger knock-on effects

on food-web structure and the biological pump[34] (see Methods for further discussion).

Taken together, our findings from the mesocosm experiments and Earth system model simulations suggest that OA will probably induce a systemic increase in $Si:N_{export}$, thereby causing a decrease in $Si(OH)_4$ in the surface ocean and a global decline of diatoms over the next centuries. This result contrasts the findings of physiological and ecological experiments that suggest that diatoms may be resistant to or benefit from OA[4–6]. Notably, the outcome in our study arises owing to a previously overlooked biogeochemical feedback in the marine silicon cycle induced by OA, not by an effect of elevated $CO_2$ on diatom physiology. Interestingly, this feedback mechanism via the silicon pump could be further exacerbated should OA favour the competitiveness of diatoms in phytoplankton communities, as indicated by aforementioned studies. In this case, diatoms would even more efficiently deplete Si in the surface ocean and accelerate their eventual decline. Our study demonstrates how expected responses of marine biota to OA can be altered or even turned upside down at the global scale through unforeseen biogeochemical feedback mechanisms. In a broader context, this exemplifies that our current understanding of biological impacts of ocean change, which is

largely based on single-species and small-scale ecological studies, might be deceptive when not considering the complexity of the Earth system.

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

## Methods

### Mesocosm experiments

**Si:N$_{export}$ measurements.** Between 2010 and 2014, we conducted five in situ mesocosm experiments to assess impacts of OA on natural plankton communities. Study sites covered a large latitudinal gradient (28 °N–79 °N) and diverse oceanic environments/ecosystems (Extended Data Fig. 1 and Extended Data Table 1). Sample collection and processing was conducted every 1 or 2 days throughout the experiments. Sinking particulate matter was obtained from sediment traps attached to the bottom of each mesocosm, thereby collecting the entire material sinking down in the enclosed water column[36]. Processing of sediment trap samples followed a previous protocol[37]. Samples for particulate matter suspended in the water column were collected with depth-integrating water samplers (HYDRO-BIOS) and filtered following standard procedures. Biogenic silica was leached from the sediment trap samples and filters by alkaline pulping (0.1 M NaOH at 85 °C). After 135 min the leaching process was terminated with 0.05 M H$_2$SO$_4$ and dissolved silica was measured spectrophotometrically[38]. Carbon and nitrogen content were determined using an elemental CN analyser (EuroEA)[39].

**Analysis of OA impacts.** To test for a systemic influence of OA on Si:N$_{export}$, we synthesized the datasets from the different experiments and (i) conducted a meta-analysis to quantify effect sizes, and (ii) computed probability density estimates. Because the experimental design, the range of CO$_2$ treatments, and the time periods for our analysis of Si:N$_{export}$ varied to some extent among experiments (Extended Data Table 1), we pooled mesocosms for ambient conditions and in the $p_{CO_2}$ range of ~700–1,000 µatm ('OA treatment'), corresponding to end-of-century values according to RCP 6.0 and 8.5 emission scenarios[15]. Effect sizes were calculated as log-transformed response ratios lnRR, an approach commonly used in meta-analysis[40]:

$$\ln(RR) = \ln(X_{OA}) - \ln(X_{control}),$$

where $X$ is the arithmetic mean of Si:N$_{export}$ ratios under OA and ambient conditions (Extended Data Table 1). Effect sizes <0 denote a negative effect of OA and effect sizes >0 indicate that the effect was positive. Effects are considered statistically significant when 95% confidence intervals (calculated from pooled standard deviations) do not overlap with zero. The overall effect size across all studies was computed by weighing individual effect sizes according to their variance, following the common methodology for meta-analyses[40]. In addition, we computed probability densities of Si:N$_{export}$ based on kernel density estimation, which better accounts for data with skewed or multimodal distributions[41]. Another advantage of this approach is that it does not require the calculation of temporal means. Instead, the entire data timeseries can be incorporated into the analysis, thus retaining information about temporal variability. Confidence intervals of the density estimates were calculated with a bootstrapping approach using data resampling (1,000 permutations)[41]. The resulting probability density plots can be interpreted analogously to histograms. Differences among ambient and OA conditions are considered statistically significant when confidence intervals of the probability density distributions do not overlap. Numbers for suspended and sinking Si, C and N (and their respective ratios) for the analysis period are given in Extended Data Table 2.

### Analysis of pH effects on Si:N in global sediment trap data

We analysed a recent compilation of global sediment trap data (674 locations collected between 1976 and 2012)[35]. The aim of this analysis was to assess the influence of pH on opal dissolution in the world ocean. In contrast to the mesocosm experiments, where export fluxes were measured only at one depth, the global dataset provides depth-resolved information, enabling us to examine the vertical change in the Si:N ratio of sinking particulate matter and how this correlates with pH. It has long been known that the silica content of sinking particles increases with depth, as opal dissolution is less efficient than organic matter remineralization[25,42]. The resulting accumulation of Si relative to N can be quantified as the change in Si:N with increasing depth, that is, the slope of the relationship of depth versus Si:N (ΔSi:N, in units of m$^{-1}$). Our approach is analogous to previous studies, which used vertical profiles of Si:C as a proxy for differential dissolution/remineralization of opal and organic matter, and its regional variability in the ocean[24,42]. We extracted all data that (I) included simultaneous measurements of Si and N, and (II) contained vertical profiles with at least three depth levels (so that ΔSi:N [m$^{-1}$] can be calculated). We then calculated linear regressions for individual Si:N profiles and subsequently extracted those for which Si:N displayed a statistically significant relationship with depth ($p < 0.05$). Thereby, profiles with no clear depth-related pattern of Si:N—for example, due water mass advection—were excluded. In total, 190 profiles of Si:N flux matched those criteria and were used for further analysis.

To assess the influence of pH on ΔSi:N, we applied linear regression analysis, using the ΔSi:N for each vertical flux profile and the average pH (from the GLODAP database)[43] over the corresponding depth range. Because temperature is considered the primary factor driving opal dissolution[16,26] and has been shown to influence the preservation of silica compared to organic matter[44], we additionally conducted a multiple regression analysis for the same dataset, by including temperature as a second factor. Results confirm that temperature also has an influence on ΔSi:N [m$^{-1}$], with a comparable effect size as identified for pH. However, because the role of temperature has been discussed extensively in earlier studies (see Methods section 'The present and future role of pH for opal dissolution in the ocean'), here we focus on the previously overlooked effect of pH.

### The present and future role of pH for opal dissolution in the ocean

It is noteworthy that pH effects on opal dissolution have so far been mostly neglected in oceanographic research and earlier work almost exclusively focused on temperature as the factor controlling opal dissolution[16,24]. Although chemical dissolution experiments have demonstrated the pH dependence of biogenic silica dissolution[20,21], it has so far been considered a minor factor in oceanic silica cycling. The reason probably lies in present-day gradients of temperature and pH in the ocean, and the resulting influence on opal dissolution: A temperature gradient of ~15–20 °C (average difference between surface and deep ocean), corresponds to a three- to fourfold change in the opal dissolution rate[26,44]. By comparison, the effect of pH described in our study is much smaller, indicating an ~20% difference in opal dissolution for present-day pH gradients of ~0.3–0.35. Accordingly, in the present ocean, the effect of temperature is roughly 10-fold larger compared to that of pH—which is probably why the latter has been so far neglected. In addition, the present-day vertical gradients of both temperature and pH both work in the same direction, that is, towards a decrease in opal dissolution rates with depth (decreasing temperature and decreasing pH).

However, the situation changes completely under future scenarios of OA and warming. Surface ocean pH is predicted to decrease by 0.2–0.4 until the year 2100, whereas sea surface warming may reach 1–3 °C (refs. [45,46]). For these changes, the effect size of both factors on opal dissolution is on a similar order of magnitude. More importantly, in contrast to present-day vertical gradients, their future changes have antagonistic effects on opal dissolution, that is, warming-driven acceleration and a pH-driven slowdown. Notably, our results suggest that the effect of decreasing pH even overcompensates for the effect of warming on a global average (Extended Data Fig. 3). This illustrates that pH becomes an increasingly relevant factor for opal dissolution and the pelagic Si cycle in the context of ongoing climate change and OA.

## Possible other contributing factors to OA impacts on Si:N

Although our results provide strong evidence for a chemical effect of decreasing pH on opal dissolution, additional or alternative explanations for the observed OA impacts on Si:N in the mesocosm studies cannot be fully excluded. On the basis of previous findings, one would expect that OA impacts on Si:N$_{export}$ can be explained by responses of diatoms. From a physiological perspective, lower pH may theoretically facilitate silicification by diatoms. The solubility of Si in seawater decreases with decreasing pH, promoting precipitation and inhibiting dissolution of opal. Diatoms are known to utilize this physicochemical property to precipitate opal in a cellular compartment with low pH conditions[47,48]. However, experimental evidence is scarce and partly controversial, with indications for either enhanced or reduced silica production under lower pH[49,50]. From an ecological perspective, higher Si:N$_{export}$ may have arisen from shifts in phytoplankton community composition, with a greater proportion of particle export driven by diatoms compared to other (non-silicifying) taxa, or by more heavily silicified species within the diatom community. However, our data do not support either of these two potential explanations, as the influence of OA on Si:N is only detectable for vertical particle fluxes (collected in sediment traps), but not for freshly produced particulate matter in the water column (Fig. 1c). This suggests that OA effects on Si:N emerged primarily while the biogenic detrital particles were sinking and not due to biotic effects during their production. Another possibility is that changes in N remineralization under simulated OA additionally contributed to the increase in Si:N. However, the current consensus is that bacterial communities and organic matter remineralization are mostly resilient to OA[51]. Results from studies that reported effects are very variable and, in most cases, it was not possible to separate direct pH effects (for example, on bacterial activity) from indirect effects mediated through pH-driven changes in quality and/or quantity of the organic matter substrate. Thus, there are currently no indications that OA will enhance N consumption of sinking organic matter. Altogether, the various independent lines of empirical evidence (mesocosms, ocean sediment traps, chemical studies) and the consistency of their results suggest that the pH effect on Si dissolution is the most probable explanation for our findings.

## Global impact assessment from Earth system model simulations

We incorporated the effects of simulated OA on Si:N$_{export}$ observed in the in situ mesocosm experiments into an Earth system model to assess the global scale impacts on nutrient availability and plankton biogeography over the coming centuries. Consequently, we applied a modified version of the University of Victoria Earth System and Climate Model (UVic ESCM), which simulates silica cycling and diatoms, as well as other functional groups of phytoplankton as described in an earlier work[52]. Biogenic opal is produced by diatoms, including a parameterization for iron dependency of silicification, resulting in elevated Si:N ratios of production under iron limitation[52]. Vertical profiles of opal fluxes and dissolved silica are instantaneously computed based on biogenic silica production in the surface ocean and dissolution throughout the water column, and silica dissolution is parameterized as an exponential, temperature-dependent rate. Simulated present-day spatial distributions of Si(OH)$_4$ in the surface ocean agree well with observational data (Extended Data Fig. 4).

To simulate OA effects on silica dissolution and Si:N$_{export}$, we parameterized the specific silica dissolution rate to scale with changes in pH throughout the water column relative to preindustrial conditions for each box in the three-dimensional model grid, thereby accounting for the vertical characteristics of future pH changes (see Extended Data Fig. 3). Therefore, we assumed that the pH sensitivity of biogenic silica dissolution derived from the observed OA effect on Si:N$_{export}$ (17% for ΔpH of around 0.3) is linear, corresponding to a decrease in opal dissolution rate of 57% per unit pH. This estimate agrees remarkably

well with published rates from chemical dissolution experiments[20,21]. We note that the model also accounts for effects of warming on silica dissolution (using a temperature dependence that is similar to other global models) that work in the opposite direction as the OA effect (see Extended Data Fig. 3).

Model simulations were run for the period 1750 to 2200 using extended IPCC scenarios (RCP 8.5 and RCP 6.0) for atmospheric CO$_2$ concentrations[27]. The reason why we conducted the simulations until the year 2200 (instead of 2100 as commonly done in other climate change studies) is that we expected impacts of the OA-driven slowdown of opal dissolution, such as Si trapping in the deep ocean, to emerge only on the long timescale of global circulation. However, as can be seen in Fig. 3b and Extended Data Fig. 5, OA-amplified Si trapping in the deep ocean and the resulting decline in diatoms become apparent by 2100. Thus, the reference year (2100 or 2200) only affects that magnitude of the effect; qualitatively, the results are very similar.

Generally, simulated impacts of climate change are consistent with other models, for example, reduced nutrient supply to the surface ocean and an associated decrease in phytoplankton biomass. The most important results that are relevant for the interpretation of OA effects reported here (via slower silica dissolution) are presented in Extended Data Table 3. More details on model behaviour in climate change simulations can be found elsewhere[52].

We emphasize that the simulated OA effect on silica dissolution occurs on top of other climate change impacts that are already known, for example, ecosystem responses to ocean warming and lower nutrient supply. Because these impacts have been extensively discussed in previous studies, we focus here on the global-scale implications of the insights from our work: the slowdown of silica dissolution under OA as revealed by our analysis of mesocosm and ocean sediment trap data. Thus, visualization and interpretation of results from the climate change simulations mostly refer to the net effect of OA-sensitive opal dissolution ($\Delta_{OA}$), which is quantified as the difference between (a) the model including OA effects on opal dissolution and (b) the standard model configuration. Additional results are presented in Extended Data Fig. 5 and Extended Data Table 3.

## Limitations of the global model

The model we used (UVic ESCM) is similar to other common models in terms of its ecosystem component, its skill in reproducing present-day conditions of biogeochemical quantities, and its behaviour in climate change simulations[52,53]. Thus, it is reasonable to assume that the simulated OA impacts on opal dissolution and Si:N$_{export}$ would yield similar results if incorporated into other global models. The driving mechanism, namely the slowdown of opal dissolution under OA, alters the vertical profile of particulate opal fluxes and regenerated Si(OH)$_4$. Because this is a chemical effect, it should be largely insensitive to specifics of the ecosystem model structure—instead, the most important factor is probably how the spatial distribution of diatoms and opal production are reproduced by different models. Our model shows good skill in reproducing observational data of Si(OH)$_4$ in the surface ocean (Extended Data Fig. 4), indicating that simulated spatial patterns of diatoms and opal production are reasonably realistic. However, as with most Earth system models, our simulations do not account for some potentially relevant mechanisms that may arise, for example, through ecological competition or complex food-web interactions, which may also yield potential repercussions for the global carbon cycle and will be discussed in the following.

In the model including OA-sensitive opal dissolution, a large part of the loss in diatom biomass is compensated by an increase in productivity of other phytoplankton groups. Accordingly, global primary productivity and carbon export remain largely unaffected by the OA effect on opal dissolution and Si:N$_{export}$, despite the sharp decline in diatoms. This is largely attributable to the degree of competition in the model, which depends on the choice of zooplankton prey

selectivity and grazing formulations[54]. By contrast, a recent modelling study focusing on functional diversity and ecological redundancy has demonstrated that changes in phytoplankton composition can entail knock-on effects on primary productivity, trophic transfer and carbon export[34]. Accordingly, it is possible that the OA-driven loss of diatoms (owing to slower opal dissolution) could trigger additional ecological changes, which might in turn modify carbon cycling and export fluxes. However, the low degree of ecological detail in most Earth system models (including ours) does not enable an assessment of such complex knock-on effects.

Another important factor to consider is iron. It is well known that iron limitation enhances silicification of diatoms, increase cellular Si:N by up to two- or threefold[55–57]. Thus, future changes in iron supply may cause shifts in the Si:N of particle flux from the surface ocean, which could theoretically counteract/enhance the consequences of OA-enhanced Si:N$_{export}$ to some extent. In this context, the largely iron-limited Southern Ocean is of key interest, as iron supply might increase in the future, owing to increased aerosol dust deposition and melting ice[45,58]. This would alleviate iron limitation and reduce the Si:N of diatoms, thereby possibly counteracting the OA-induced increase in Si:N$_{export}$. Our model includes a parameterization for iron-limitation effects on opal production by diatoms, thereby also controlling Si:N$_{export}$. However, as with most other Earth system models[59], it does not account for the complex mechanisms that may lead to future changes in iron deposition (for example, aeolian dust deposition, input from ice melting). Simulated iron inputs to the ocean are fixed at preindustrial rates. Thus, it is not possible to directly assess how future changes in iron limitation might interact with OA-driven changes in opal dissolution. However, in this context it is important to differentiate between effects on Si:N in the surface ocean (for example, iron effects on during production) and those occurring throughout the water column (for example, due to OA effects on dissolution): changes in the total amount of produced biogenic silica would alter the magnitude of Si flux from the euphotic zone, whereas the pH effect on Si dissolution alters flux attenuation throughout the water column. The relative size of the OA effect (that is, the proportional decrease in dissolution of sinking opal) is thus independent of the magnitude of Si flux from the surface. Accordingly, it can be assumed that future changes in dust deposition and Si:N (for example, in the Southern Ocean) would be superimposed by the OA-driven decrease in opal dissolution.

Furthermore, effects of opal ballasting on sinking velocities and remineralization rates of in diatom-derived organic matter are not accounted for in the model. In theory, the OA-driven decrease in opal dissolution may increase particle sinking velocities owing to enhanced (that is, prolonged) mineral ballasting by opal[60]. At the same time, slower opal dissolution may enhance the protection of organic matter against remineralization to some extent[61]. On the basis of such considerations, it is often suggested that a decrease in diatoms (such as the OA-driven loss of diatoms reported here) may reduce the efficiency of the biological carbon pump owing to slower sinking speed and/or faster remineralization of sinking organic matter of non-diatom origin[1]. However, as the mechanisms on the particle scale described above are very complex and evidence on mineral ballasting in observational data is controversial, they are not included in our model (and neither in other, ecologically more complex, models)[34,62]. Nevertheless, these mechanisms should be kept in mind when interpreting our results, as they could potentially have repercussions for the efficiency of the biological carbon pump.

Altogether, we note that our model results should be considered as an important estimate of how OA impacts on opal dissolution and Si:N$_{export}$ may affect the marine silica cycle and the future biogeography of diatoms. As the OA-driven decrease in opal dissolution is a purely abiotic mechanism, we consider the main findings from our model simulations (decline in Si(OH)$_4$ and diatoms) to be robust and valid on

large spatiotemporal scales, and expect them to be reproducible with other Earth system models. However, the model properties outlined above result in only minor knock-on effects of OA-driven changes on primary productivity and carbon export. Whether this holds true for the real ocean is uncertain and other Earth system models, for example, those with a higher degree of ecological complexity, may yield somewhat different results. For instance, although the OA effect on Si(OH)$_4$ and diatoms might be similar, possible responses of plankton community structure and biogeochemical processes may differ depending on the ecosystem model structure. Therefore, we hope that our study will be an incentive for the scientific community to explore OA effects on opal dissolution with different global models, and thereby assessing the potential variability of this effect among models.

## Data availability

The data from the mesocosm experiments used for this study are provided as Supplementary Data, and are also archived on the PANGAEA database (https://doi.org/10.1594/PANGAEA.940756). Furthermore, additional data from the individual mesocosm experiments can be found on the PANGAEA database using the keyword 'KOSMOS'.

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

**Acknowledgements** This study was supported by the German Federal Ministry of Science and Education (BMBF) in the framework of the projects BIOACID III ('Biological Impacts of Ocean Acidification', FKZ 03F0728) and SOPRAN III ('Surface Ocean Processes in the Anthropocene', FKZ 03F0662). Funding for the different mesocosm studies was also provided by the 'European Project on Ocean Acidification' (EPOCA, grant no. 211384) from the European Community's Seventh Framework Programme (FP7/2007–2013), the EU project MESOAQUA (grant no. 228224), as well as BIOACID II (FKZ 03F06550) and SOPRAN II (03F0611). Furthermore, we thank all participating scientists and technicians for their efforts in realizing the different studies. We also thank the staff of the marine biological stations in the different study locations for providing logistics, technical assistance and support at all times. We thank the captains and crews of RV *Viking Explorer*, MV *Esperanza* of Greenpeace, RV *Håkon Mosby* (2011609), RV *Alkor* (AL376, AL394, AL397, AL406, AL420), RV *Heincke* (HE360), RV *Poseidon* (POS463), and RV *Hesperides* (29HE20140924) for support during transport, deployment and recovery of the mesocosm facilities.

**Author contributions** This study was conceived by J.T. Mesocosm experiments were coordinated and implemented by U.R. with support and data acquisition by T.B., L.T.B. and J.T. Analysis of global sediment trap data was conducted by J.T. Model implementation was realized by K.K., A.E.F.P. and J.T. Analysis and visualization of model results was done by J.T. The manuscript was written by J.T. with contributions from all co-authors.

**Funding** Open access funding provided by GEOMAR Helmholtz-Zentrum für Ozeanforschung Kiel.

**Competing interests** The authors declare no competing interests.

**Additional information**
**Correspondence and requests for materials** should be addressed to Jan Taucher.

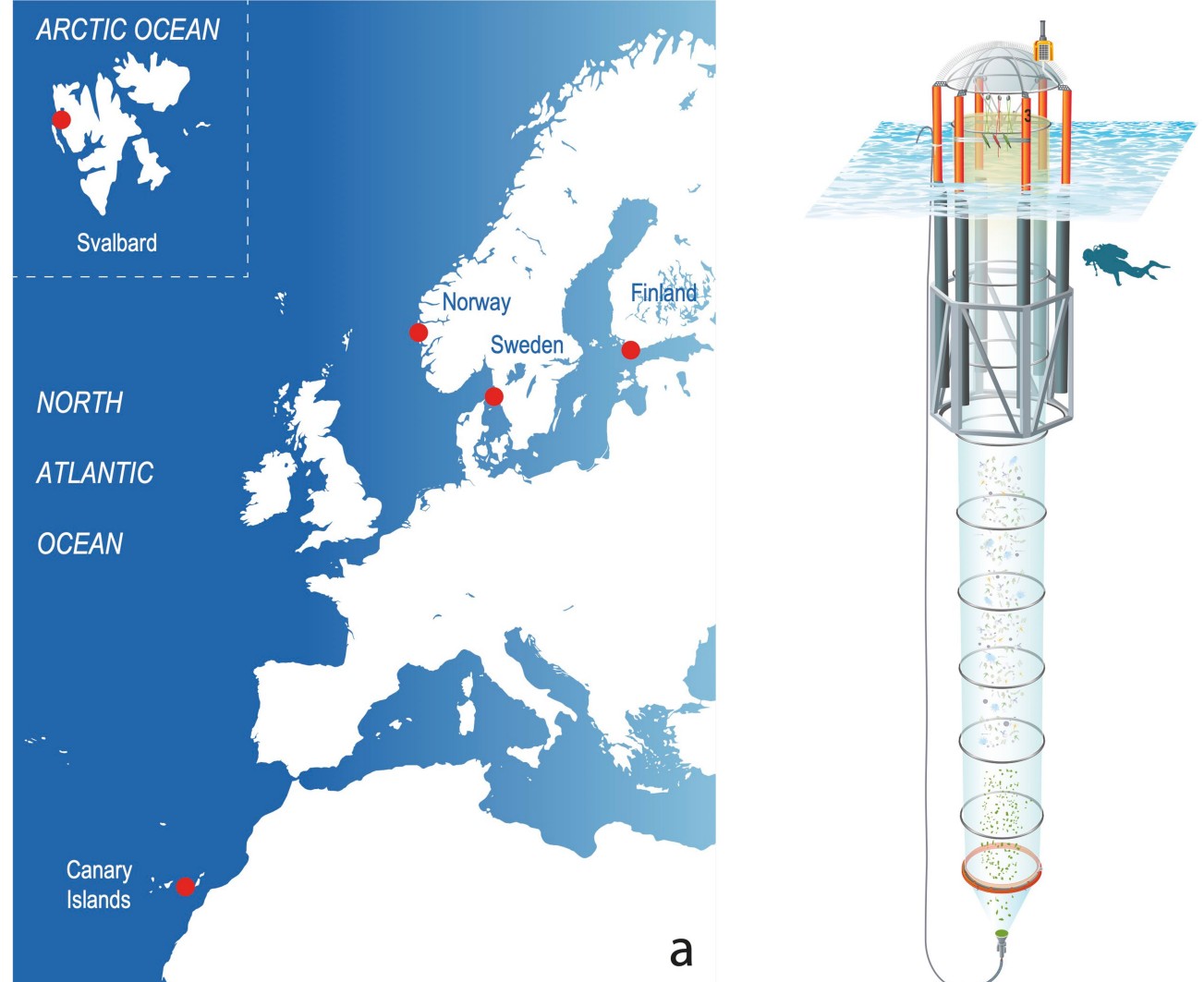

a

b

**Extended Data Fig. 1 | Mesocosm experiments on OA. a**, Locations of in situ mesocosm experiments with the 'Kiel Off-Shore Mesocosms for Ocean Simulations' (KOSMOS). **b**, Schematic drawing of a pelagic mesocosm enclosing the natural plankton community and collecting sinking organic matter in a full-diameter sediment trap at the bottom (15 to 25 m water depth, depending on the study site). Different OA scenarios were achieved by adding filtered, $CO_2$-saturated seawater equally distributed into the mesocosms[36] at the beginning of the experiment, and usually several more times throughout the study period to maintain $p_{CO_2}$ within target levels. Illustration by R. Erven and T.B. (GEOMAR), reprinted with permission from the AGU and Springer Nature, ref.[63].

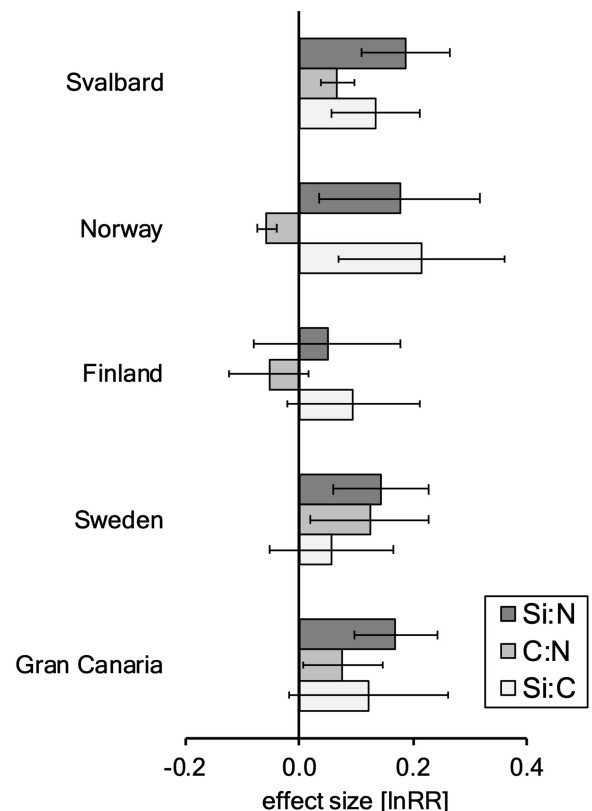

**Extended Data Fig. 2 | Comparison of OA impacts on different elemental ratios (Si:N, C:N and Si:C) of sinking biogenic matter.** Note that effects on Si:C display a larger variability than Si:N owing to additional (but variable) responses of C:N ratios. In contrast to the OA effect on Si (slower chemical dissolution), the shifts in C:N are biotically driven, resulting from responses of primary producers and heterotrophic consumers, which together leave an imprint on the C:N sinking out of the productive surface layer [63]. Depending on the direction and magnitude of the OA effect on C:N, this offsets or amplifies the effect on Si:C compared to Si:N. However, because OA effects on C:N are driven by biotic processes and linked to freshly produced material, it can be assumed that this effect is mostly restricted to the surface ocean, whereas the OA effect on Si occurs throughout the entire water column (see Extended Data Fig. 3). It can thus be expected that the OA-driven decrease in opal dissolution will enhance the preservation of Si compared to C, thereby increase Si:C ratios of particles sinking through the water column.

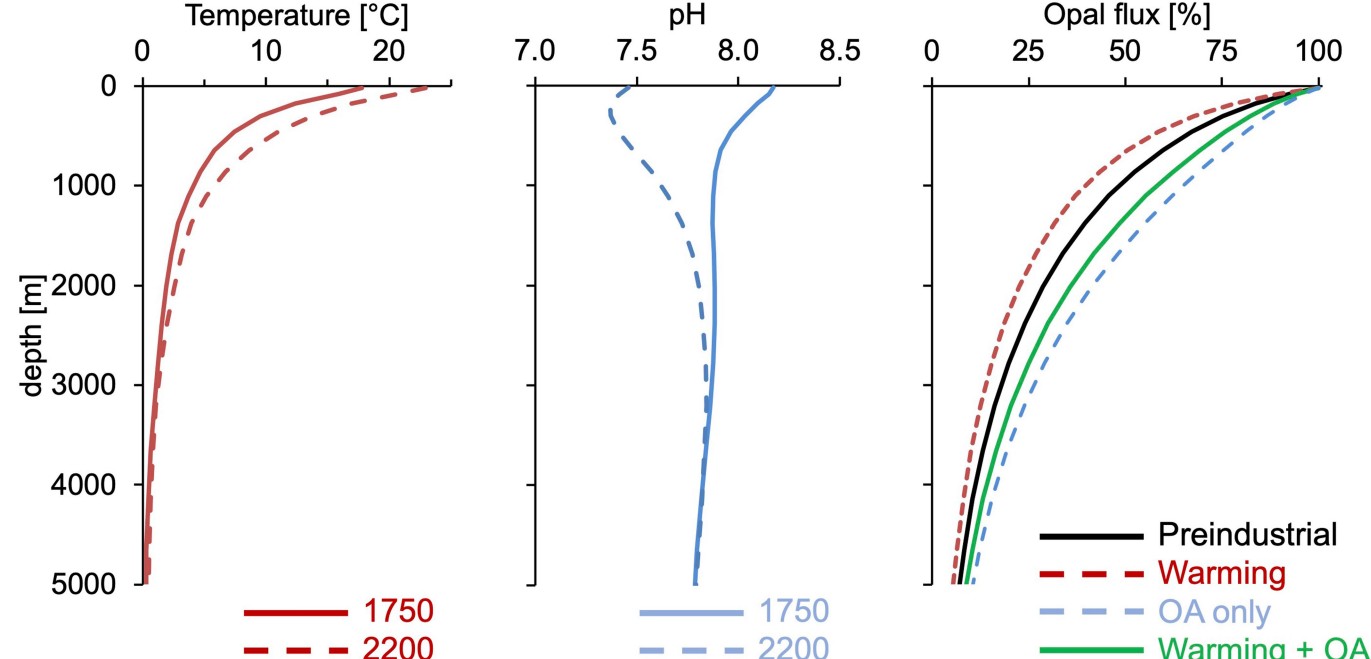

**Extended Data Fig. 3 | Comparison of warming and OA impacts on silica dissolution.** One-dimensional representation of warming and OA effects on silica dissolution and export to depth in our model, using globally averaged vertical profiles of temperature (left) and pH (middle) under preindustrial conditions and for the year 2200 (values from the RCP8.5 simulation). The right panel depicts the model parameterization for opal dissolution, applied to the corresponding temperature and pH profiles. Averaged over the upper 1,000 m, the specific opal dissolution rate ($Diss_{opal}$) under preindustrial conditions amounts to 0.0008 (in units of $m^{-1}$), that is, 0.08% of sinking opal is dissolved per metre, resulting in a characteristic 'Martin curve' of opal flux attenuation with depth (black line). $Diss_{opal}$ in the model is temperature-dependent and roughly doubles for an increase in temperature of 10 °C. Warming of ~3 °C in the upper ocean until the year 2200 (averaged over the upper 1,000 m) increases $Diss_{opal}$ by about 1.3-fold (from 0.08% to 0.11%), thereby leading to a stronger flux attenuation with depth (red dashed line) compared to preindustrial conditions. At the same time, pH decreases by around 0.6 units until year 2200 (averaged over the upper 1,000 m). According to our findings, opal dissolution rates are 57% lower per unit decrease in pH, (that is, a decrease by a factor of ~2.5). This effect of pH works in the opposite direction as the temperature effect, that is, slowing down opal dissolution and thus weakening the decline of opal with depth. When considering this pH effect in isolation in our example (that is, not accounting for the effects of warming; dashed blue line), this drop in pH would decrease $Diss_{opal}$ from 0.08% to 0.045% (that is, reducing it by a factor of 1.8). When considering both warming and acidification at the same time, the pH effect is still strong enough to over-compensate the temperature effect. $Diss_{opal}$ amounts to 0.059%, which is notably lower than under preindustrial (0.08%) and warming-only (0.11%) conditions. Accordingly, opal dissolution in a warmer and lower-pH ocean is slower than under preindustrial conditions, thereby leading to an enhanced efficiency of silica export to the deep ocean (green line).

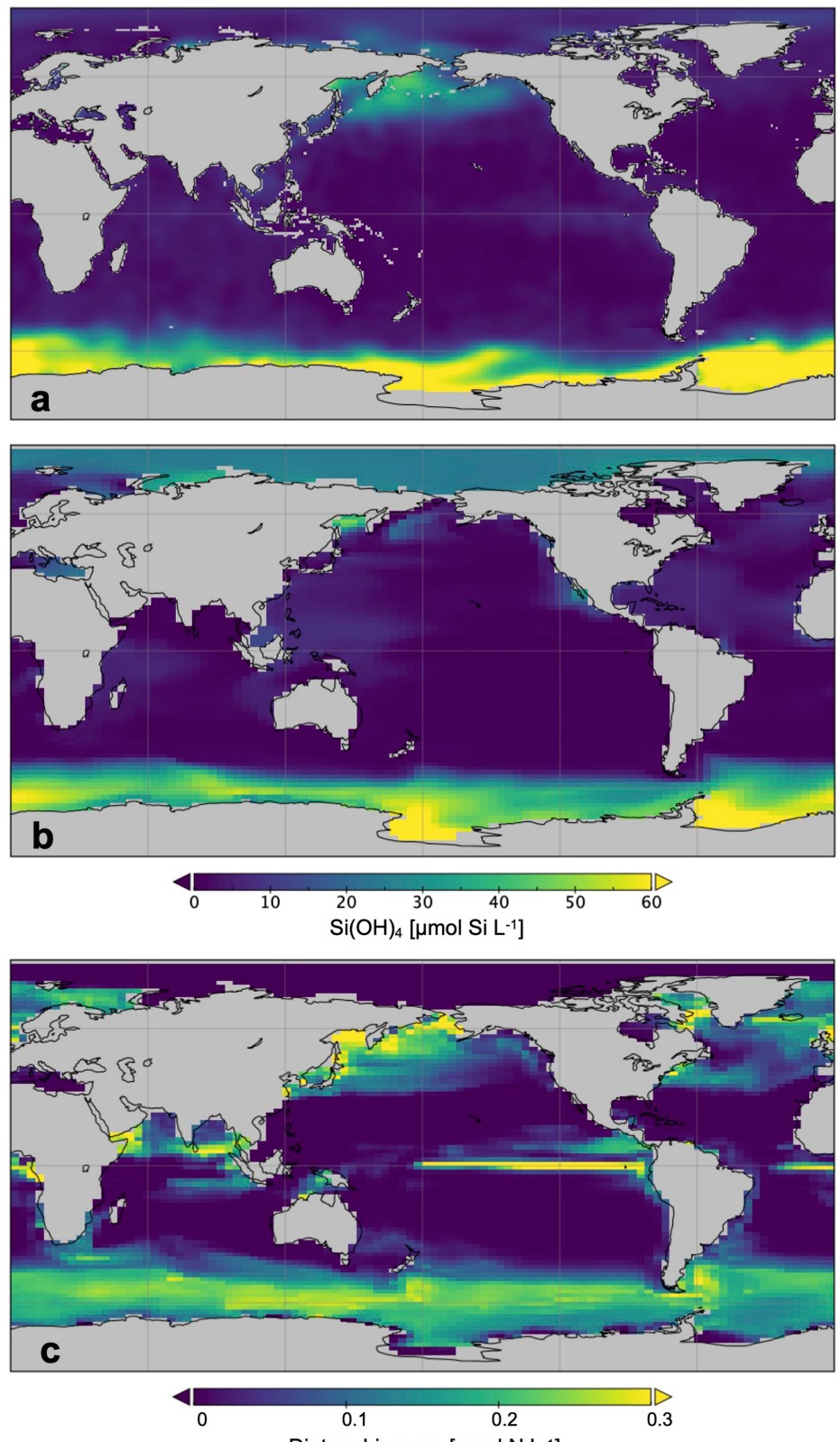

**Extended Data Fig. 4 | Model versus data for silicic acid in the surface ocean. a**, **b**, Present-day distribution of sea surface $Si(OH)_4$ from the World Ocean Atlas[64] (**a**) and simulated by our model (**b**). **c**, Simulated distribution of diatom biomass.

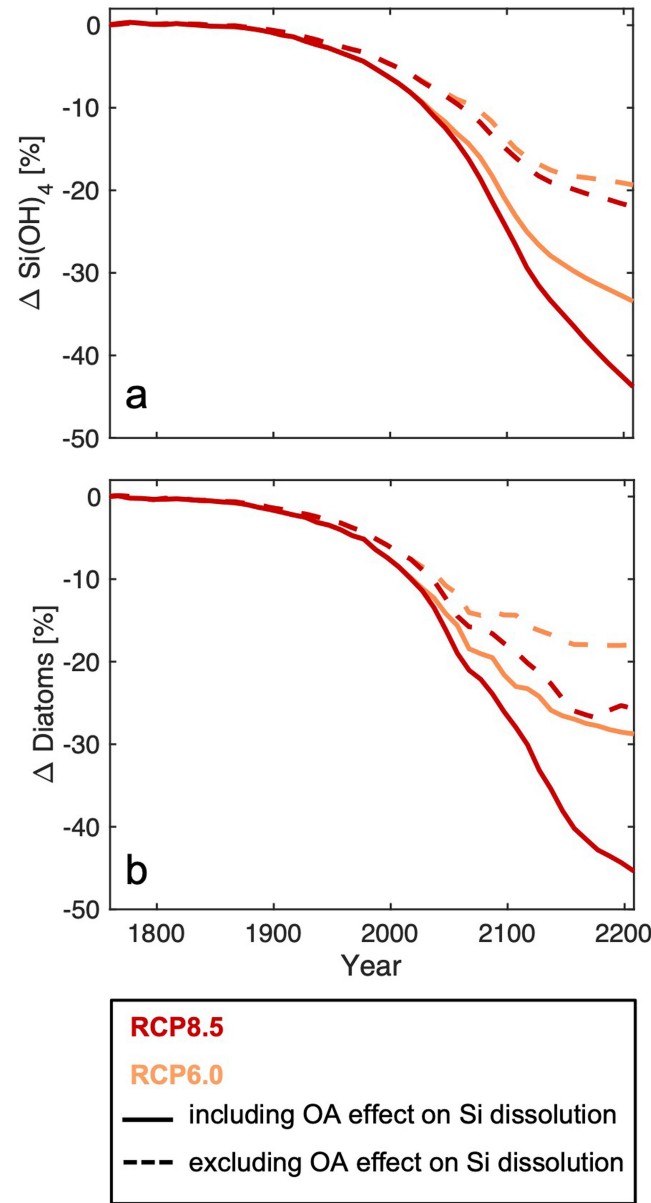

**Extended Data Fig. 5 | Simulated OA impacts on silicic acid and diatoms.**
**a**, **b**, Future changes in Si(OH)$_4$ (**a**) and diatom biomass (**b**) in the surface ocean
relative to preindustrial conditions. Shown are results from simulations with
the standard model configuration, that is, excluding OA-effects on silica
dissolution (dashed lines), and the model including OA effects on silica
dissolution (solid lines) for RCP6.0 and RCP8.5 emission scenarios.
See Methods and Extended Data Table 3 for more details.

**Extended Data Table 1 | Overview of study locations, experimental design and environmental conditions of the different mesocosm experiments**

| Location | Svalbard | Norway | Finland | Sweden | Gran Canaria |
|---|---|---|---|---|---|
| Latitude | 79°N | 60°N | 60°N | 58°N | 28°N |
| Biome | Arctic | Temperate | Temperate | Temperate | Subtropical |
| Year | 2010 | 2011 | 2012 | 2013 | 2014 |
| Month | June - July | May - June | June - August | March - June | September - December |
| Duration of experiment [days] | 30 | 34 | 43 | 103 | 55 |
| Period for analysis of $Si:N_{export}$ | day 1–22 | day 16–34 | day 7–43 | day 1–103 | day 1–25 |
| Seasonal cycle / trophic state | Early summer post-bloom | Late spring bloom | post-bloom nutr.-depleted | entire spring bloom | oligotrophic |
| Experiment design | gradient | gradient | gradient | replicated treatments | gradient |
| Number of mesocosms | 9 | 8 | 6 | 10 | 8 |
| $pCO_2$ range total [µatm] | 190 – 1000 | 310 – 1600 | 350 – 1120 | 380 – 760 | 400 – 1050 |
| "Ambient" $pCO_2$ [µatm] | 210 ± 20 (n=3) | 340 ± 30 (n=2) | 350 ± 5 (n=2) | 380 ± 20 (n=5) | 425 ± 30 (n=3) |
| "OA" $pCO_2$ [µatm] | 850 ± 150 (n=3) | 900 ± 90 (n=3) | 840 ± 80 (n=2) | 760 ± 20 (n=5) | 940 ± 90 (n=3) |
| Salinity | 34 | 32 | 6 | 29 | 37 |
| Temperature range [°C] | 2.0 – 5.5 | 7.9 – 10.1 | 8.0 – 12.1 | 0.5 – 16.0 | 22.2 – 24.3 |
| Nutrient range [µmol $NO_3^-$ $L^{-1}$] | <0.1 – 5.5 | <0.1 – 5.3 | <0.01 – 0.05 | <0.1 – 7.0 | <0.1 |
| Chl $a$ range [µg $L^{-1}$] | 0.2 – 3.0 | 1.0 – 6.0 | 1.1 – 2.2 | 0.5 – 5.5 | <0.1 |
| Mesocosm depth [m] | 15 | 25 | 19 | 19 | 15 |
| Mesocosm volume [$m^3$] | 45 | 75 | 55 | 50 | 35 |
| Reference | Schulz et al.[65] | Riebesell et al.[66] | Paul et al.[67] | Bach et al.[68] | Taucher et al.[69] |

Note that some experiments covered larger $pCO_2$ ranges than considered in the analysis of OA effect sizes in Fig. 1, because we have limited the 'future OA' scenario to $pCO_2$ levels between ~700 to 1,000 µatm (see Methods). We note that the ambient treatment comprises mesocosms without $CO_2$ perturbation, and in cases of gradient setups, those with a $CO_2$ elevated by a maximum of 100 µatm relative to ambient. Specified $pCO_2$ values represent the average over the respective study period. If possible, the analysis of $Si:N_{export}$ comprised the entire experimental duration. However, some time periods had to be excluded from this analysis, if temporal dynamics of plankton succession and particle export did not allow for an accurate analysis of $pCO_2$ effects. This was the case when (1) the sedimentation of a preceding plankton bloom had not reached the sediment traps or was still ongoing at the end of the study (Svalbard after day 22, Gran Canaria after day 25), or (2) export flux originated from biomass produced during the initial experimental period, before $pCO_2$ manipulation was completed (Finland days 1–5 and Norway days 1–15). Refs. [65–69].

**Extended Data Table 2 | Mass fluxes of Si, N and C and their elemental ratios**

| Location | $pCO_2$ [µatm] | Si flux [µmol L$^{-1}$ d$^{-1}$] | N flux [µmol L$^{-1}$ d$^{-1}$] | C flux [µmol L$^{-1}$ d$^{-1}$] | Si:N | Si:C | C:N |
|---|---|---|---|---|---|---|---|
| Svalbard | 568 | 0.004 | 0.042 | 0.277 | 0.099 | 0.015 | 6.578 |
| Svalbard | 245 | 0.003 | 0.041 | 0.240 | 0.081 | 0.013 | 6.092 |
| Svalbard | 177 | 0.004 | 0.051 | 0.319 | 0.080 | 0.013 | 6.299 |
| Svalbard | 324 | 0.003 | 0.044 | 0.261 | 0.084 | 0.013 | 6.211 |
| Svalbard | 800 | 0.004 | 0.043 | 0.284 | 0.095 | 0.014 | 6.716 |
| Svalbard | 659 | 0.004 | 0.044 | 0.276 | 0.089 | 0.014 | 6.491 |
| Svalbard | 178 | 0.003 | 0.037 | 0.225 | 0.074 | 0.012 | 6.251 |
| Svalbard | 405 | 0.003 | 0.039 | 0.241 | 0.084 | 0.013 | 6.523 |
| Svalbard | 999 | 0.004 | 0.039 | 0.257 | 0.099 | 0.015 | 6.723 |
| Norway | 644 | 0.034 | 0.051 | 0.437 | 0.696 | 0.081 | 8.550 |
| Norway | 773 | 0.037 | 0.061 | 0.495 | 0.692 | 0.081 | 8.306 |
| Norway | 305 | 0.035 | 0.066 | 0.592 | 0.591 | 0.066 | 8.943 |
| Norway | 868 | 0.042 | 0.059 | 0.507 | 0.779 | 0.089 | 8.480 |
| Norway | 365 | 0.035 | 0.057 | 0.499 | 0.679 | 0.076 | 8.862 |
| Norway | 1127 | 0.038 | 0.052 | 0.439 | 0.803 | 0.093 | 8.451 |
| Norway | 491 | 0.032 | 0.045 | 0.382 | 0.765 | 0.088 | 8.547 |
| Norway | 1476 | 0.030 | 0.044 | 0.348 | 0.813 | 0.097 | 8.099 |
| Finland | 350 | 0.042 | 0.057 | 0.439 | 0.606 | 0.079 | 7.392 |
| Finland | 919 | 0.038 | 0.054 | 0.387 | 0.602 | 0.083 | 6.809 |
| Finland | 354 | 0.033 | 0.051 | 0.359 | 0.537 | 0.075 | 6.682 |
| Finland | 754 | 0.034 | 0.055 | 0.387 | 0.531 | 0.075 | 6.654 |
| Finland | 468 | 0.030 | 0.040 | 0.313 | 0.521 | 0.067 | 7.382 |
| Finland | 1117 | 0.040 | 0.057 | 0.411 | 0.616 | 0.084 | 6.890 |
| Sweden | 365 | 0.217 | 0.126 | 1.481 | 1.769 | 0.169 | 11.592 |
| Sweden | 764 | 0.204 | 0.101 | 1.148 | 2.144 | 0.201 | 11.430 |
| Sweden | 399 | 0.195 | 0.121 | 1.214 | 1.631 | 0.164 | 10.362 |
| Sweden | 748 | 0.207 | 0.126 | 1.664 | 2.009 | 0.173 | 13.177 |
| Sweden | 405 | 0.201 | 0.119 | 1.101 | 1.751 | 0.192 | 9.390 |
| Sweden | 757 | 0.212 | 0.109 | 1.387 | 2.100 | 0.181 | 12.801 |
| Sweden | 769 | 0.177 | 0.102 | 1.247 | 1.856 | 0.167 | 12.025 |
| Sweden | 778 | 0.200 | 0.108 | 1.162 | 2.003 | 0.204 | 10.588 |
| Sweden | 390 | 0.184 | 0.110 | 1.191 | 1.981 | 0.191 | 10.654 |
| Sweden | 361 | 0.204 | 0.139 | 1.607 | 1.627 | 0.160 | 11.034 |
| Gran Can. | 401 | 0.011 | 0.021 | 0.206 | 0.498 | 0.052 | 9.357 |
| Gran Can. | 924 | 0.007 | 0.013 | 0.174 | 0.593 | 0.057 | 10.927 |
| Gran Can. | 588 | 0.013 | 0.022 | 0.243 | 0.583 | 0.054 | 10.380 |
| Gran Can. | 724 | 0.012 | 0.023 | 0.228 | 0.506 | 0.052 | 9.402 |
| Gran Can. | 481 | 0.013 | 0.029 | 0.309 | 0.460 | 0.047 | 9.525 |
| Gran Can. | 863 | 0.012 | 0.022 | 0.234 | 0.544 | 0.052 | 10.258 |
| Gran Can. | 679 | 0.013 | 0.028 | 0.286 | 0.490 | 0.048 | 9.745 |
| Gran Can. | 1047 | 0.014 | 0.026 | 0.285 | 0.552 | 0.051 | 10.301 |
| Gran Can. | 401 | 0.018 | 0.036 | 0.408 | 0.468 | 0.042 | 10.295 |

Data shown are export flux of different elements for the different study locations and $pCO_2$ levels, given as temporal averages over the analysis period. Note that elemental ratios given here are calculated from the entire data timeseries of each mesocosm (and not the temporal averages of Si, N and C shown here), in order to preserve the temporal variability of the raw data. The full timeseries raw data are available as Supplementary Data, as well as in the PANGAEA data archive (see 'Data availability').

**Extended Data Table 3 | Summary of results from model simulations**

| | Year | Si(OH)$_4$ | NO$_3$ | Diatoms | Phyto$_{small}$ |
|---|---|---|---|---|---|
| **RCP6.0** | | | | | |
| *Change relative to preindustrial conditions (year 1750)* | | | | | |
| Standard model (no OA impacts on Si) | 2100 | -13% | -8% | -14% | -2% |
| | 2200 | -19% | -10% | -18% | -1% |
| Including OA impacts on Si dissolution | 2100 | -21% | -8% | -22% | +5% |
| | 2200 | -33% | -10% | -29% | +9% |
| *Net OA effect (including vs. excluding OA impacts on Si dissolution)* | | | | | |
| Net OA effect ($\Delta_{OA}$) | 2100 | -9% | 0% | -8% | +7% |
| | 2200 | -17% | 0% | -13% | +10% |
| **RCP8.5** | | | | | |
| *Change relative to preindustrial conditions (year 1750)* | | | | | |
| Standard model (no OA impacts on Si) | 2100 | -15% | -10% | -18% | -2% |
| | 2200 | -22% | -13% | -25% | 0% |
| Including OA impacts on Si dissolution | 2100 | -24% | -10% | -26% | +6% |
| | 2200 | -42% | -12% | -45% | +19% |
| *Net OA effect (including vs. excluding OA impacts on Si dissolution)* | | | | | |
| Net OA effect ($\Delta_{OA}$) | 2100 | -11% | 0% | -10% | +8% |
| | 2200 | -27% | 0% | -26% | +18% |

Comparison of global-scale impacts of OA-driven slowdown of opal dissolution versus other climate change impacts in our simulations with the UVic ESCM model, given for the key quantities of our study: Si(OH)$_4$, NO$_3$ as well as biomass of diatoms and small phytoplankton (Phyto$_{small}$) in the surface ocean. Shown are temporal changes relative to preindustrial conditions in the standard model (excluding OA-effects on Si dissolution) and the model including OA effects on Si dissolution, as well as the net OA effect given as the difference between the two model configurations ($\Delta_{OA}$, see Methods). Simulated impacts of climate change are consistent with other models, predicting reduced nutrient supply to the surface ocean, owing to enhanced thermal stratification[30,31,45], a concomitant long-term nutrient trapping in the deep ocean[70], as well as a resulting decline in phytoplankton biomass (mostly diatoms) and shift towards low-nutrient adapted ('small') phytoplankton[71–73]. In the simulations including OA effects on Si dissolution, the loss of Si(OH)$_4$ from the surface ocean is strongly amplified, almost doubling the long-term silica trapping in the deep ocean (for example, from −22% to −42% in 2200 under RCP8.5). This additional loss of Si(OH)$_4$ from the surface ocean exacerbates nutrient limitation for diatoms, thereby amplifying their decline under climate change by up to 1.8-fold (for example, from −25% to −45% in 2200 under RCP8.5). See Extended Data Fig. 5 for a visualization of results for Si(OH)$_4$ and diatom biomass.