## [Peer Review File · Nature]

Manuscript Title: Enhanced silica export in a future ocean triggers global diatom decline

Reviewer Comments & Author Rebuttals

Reviewer Reports on the Initial Version:

Referees' comments:

Referee #1 (Remarks to the Author):

Review : Global decline of diatoms through ocean acidification » by Taucher and co-authors

Taucher and co-authors use (1) five in situ mesocosm experiments, and (2) a compilation of sediment trap data to demonstrate that ocean acidification increases the elemental ratio of Si (silicon) to N (nitrogen) in sinking biogenic material. They attribute this change in Si/N ratio to the slower chemical dissolution of biogenic silica when seawater pH is lower.

They then use an earth system model of intermediate complexity (UVIC ESM) to test the significance of this potential effect (a pH-induced change in silica dissolution) in future projections. While diatoms have been shown to be resistant to, and even benefit from, ocean acidification, their model results suggest that the increase in Si/N ratio in response to future ocean acidification would result in the trapping of silicon in the deep ocean, to a more silicon-depleted surface ocean, and thus trigger an overall decline in diatom abundance in the long-term future ocean.

I am in principle supportive, both of the results the authors present in their manuscript and of the methods they used, combining mesocosm experiments, a compilation of sediment trap data, and some specific model simulations. The manuscript is well written and easy to follow. The figures are of good quality.

Their main result on the decrease in diatom abundance with ocean acidification is clearly at odds with the past literature and illustrates well how long-term time scales of ocean circulation superimposed on geochemical changes can redistribute nutrients between the surface and deep oceans and lead to counter-intuitive responses (ie. a decline in diatom abundance, whereas direct acidification effects may indeed be beneficial to diatoms).

That being said, I have some major reservations that prevent me from recommending this article for publication in Nature, at least in its current state. Because of my personal expertise, I'm mostly commenting here on the modelling aspects of the manuscript.

(1) On the choice to focus the model output analysis on the long-term (2200 instead of the end of the 21st century).

Most of the model projections that have been analysed and published so far in terms of the response of phytoplankton distribution to anthropogenic climate change focus on the end of the century (typically 2090-2100 or 2080-2100). These model studies suggest that the shifts in communities from projected reduction of nutrient supplies do favour smaller phytoplankton species over larger types such as diatoms (e.g. Bopp et al., 2005, Marinov et al. 2010, Dutkiewicz et al., 2013, Dutkiewicz et al. 2015).

Here, the authors have chosen to specifically focus their analysis on the simulated response at the

end of the 22nd century – thus preventing a clear comparison to the above-mentioned model studies. I understand their choice is somehow dictated by the fact that their simulated response is linked to the long time-scale of the ocean circulation. But a much clearer justification of this choice is needed. I would also suggest to detail the time-scale of the response they obtain.

(2) On the choice to focus the model output analysis on the response to Ocean Acidification only. The impact of an acidification-induced reduction in biogenic silica dissolution is indeed well illustrated by showing only differences between 2 model simulations (with / without the impact of pH changes on dissolution). But this leaves many questions unanswered, which at least merit some quantitative analysis and some discussion in the main text of the manuscript. What is the relative effect of decreasing pH vs. increasing temperature on opal dissolution and the subsequent biogeochemical impacts? How does the silica trapping effect (due to Si/N changes) compare to the other effects that have been discussed in the literature (due to the direct effects of ocean acidification on calcifying plankton types, or to the nutrient-induced shift towards smaller phytoplankton types?). In that regard, the supplementary discussion, addressing some of these questions, is much too qualitative and would benefit from a more thorough analysis.

(3) On the incorporation of the pH effect on biogenic silica dissolution

Finally, I have some reservations about how the effect of pH on dissolution, which is a key component of this study, is incorporated! As I understand it, biogenic silica export and associated dissolution in the water column is represented implicitly - with "vertical profiles of dissolved silica in the water column are instantaneously calculated based on biogenic silica production in the surface ocean." And silica dissolution is scaled "with pH changes from pre-industrial conditions for each model grid cell" in the new modified version. Are pH changes in the water column accounted for? Or only the pH changes at the ocean surface? This would be problematic and lead to an overestimation of the effect studied, as the pH changes at depth are likely to be much smaller than those simulated in the surface ocean.

Simply put, I found the model description to be far too light and not consistent enough. The authors need to do a better job of detailing the new parameterization.

Bopp, L. et al. (2005). Response of diatoms distribution to global warming and potential implications: A global model study. *Geophysical Research Letters* 32.

Dutkiewicz, S., Scott, J.R., and Follows, M.J. (2013). Winners and losers: Ecological and biogeochemical changes in a warming ocean. *Global Biogeochemical Cycles* 27, 463–477.

Dutkiewicz, S., Morris, J.J., Follows, M.J., Scott, J., Levitan, O., Dyhrman, S.T., and Berman-Frank, I. (2015). Impact of ocean acidification on the structure of future phytoplankton communities. *Nature Climate Change* 5, 1002–1006.

Marinov, I., Doney, S.C., and Lima, I.D. (2010). Response of ocean phytoplankton community structure to climate change over the 21st century: partitioning the effects of nutrients, temperature and light. *Biogeosciences* 7, 3941–3959.

Referee #2 (Remarks to the Author):

Riebesell and colleagues provide evidence for slower silica dissolution with reduced pH, which leads to higher Si:N ratios in particulate matter sinking to depth. Using a Global Climate Model, the authors forecast how this abiotic change in dissolution affects the availability of Si in the surface ocean - resulting in a decline in diatom productivity in the future ocean.

This research is novel and challenges the current interpretation of the primary literature surrounding the effect of ocean acidification on diatoms - which up to this point, generally concludes that ocean acidification will benefit diatoms through physiological or ecological mechanisms.

The authors use appropriate analyses for the meta-analysis and global analysis of sediment trap data. I am less familiar with the methods used in GCMs and suggest including a reviewer that frequently uses these models in their research. I was a little unclear about how to compare the confidence intervals for the probability density kernels, given they all overlap for some portion of the distributions. Some clarifications could be helpful here.

While the results are novel and note-worthy, several of the points brought up in the current supplementary discussion deserve more discussion in the primary text. Most interesting to me is the limitations of the model used - such as how changes in the biology of the ecosystem could enhance or reduce the effects detected here. I would like to see how this mechanism plays out in other GCMs, given the results and conclusions are likely to be heavily influenced by the specifics of the model used. Model comparison would strengthen the findings of the forecasts.

Overall, this is an extremely important and interesting study that highlights the complexity of global effects of ocean acidification and climate change on marine ecosystems. While there are some limitations in how much should be concluded from a single GCM, I think that the data are compelling enough to warrant publication and discussion.

Referee #3 (Remarks to the Author):

This manuscript presents a meta-analysis of five mesocosm experiments that examined the effects of ocean acidification on Si:N export ratios in the ocean. The authors are world experts in doing these types of large volume incubations, and the experiments and data analyses seem carefully and correctly done. Their major finding that exported Si:N decreases by 17% under OA conditions in four out of five of their experiments is surprisingly consistent, especially considering that their statistical methodology required them to average consolidated results from mesocosms with fairly widely varying CO₂ concentrations (lines 274-277). They make a good case that this OA effect could be a feedback mechanism not yet considered by biogeochemists. However, they also could do a much better job of recognizing the importance of other equally or more impactful environmental feedbacks on Si:N export, such as those arising from iron limitation and warming (see my comments below). I have four major comments, as well as some minor corrections and suggestions.

My first major comment is that readers need to have access to the concentrations of particulate Si and N in the mesocosms and trap samples, not just the ratios. Obviously, a change in a ratio can be due to changes in either the numerator or the denominator, or to both. There is no way to tell whether their explanation that Si is dissolving more slowly at lower pH values is correct- it could just as easily be that N is remineralizing faster, or both could be happening at once. Please make available the raw data from which the ratios are calculated, and discuss the changes in the absolute fluxes of Si and N from the suspended plankton to the trap collections in both the mesocosms and (where possible) the meta-analyses of in situ sediment traps- not just the ratios.

A similar issue is the need to look for additional support for this Si dissolution trend using a different normalizer. Please plot the Si:C ratios as well as the Si:N ratios, and their changes from the suspended particulate matter to the trap collections (and the raw POC data they are based on, as above). These POC data should be easily available, since they did standard CN analyses to obtain the PON values. Considering Si:C ratios also would allow the authors to link Si fluxes directly to carbon export, something their ES model is not able to do (see lines 600-604 in the SI). A discussion of coupling to carbon export, or lack thereof, would greatly expand the biogeochemical and climate relevance of their take home message, which now focuses exclusively on Si export.

The outcome of their model experiment suggests OA effects on Si dissolution in the Southern Ocean could have a major global effect on Si availability to diatoms through downstream reductions in upwelled Si at lower latitudes. This scenario has previously been raised a number of times in the literature for iron limitation, for instance in the Brzezinski et al. 2002 paper already cited in the text. This is because iron limitation greatly increases the Si:N and Si:C ratios of diatoms, by up to 200-300% in some cases (see older papers by Hutchins, Takeda and Brzezinski). Given that most models predict increases in iron supplies to the Southern Ocean in the future from a dustier climate and melting ice (see recent papers by Mahowald, for instance), isn't it possible that the relatively modest 17% reduction in sinking Si:N they model may be overwhelmed by much larger Si:N changes in diatoms due to iron limitation shifts? At the very least, a robust caveat is needed for the modeling component, as it apparently lacks dynamic terms for either changes in iron supply or their consequences for diatom Si:N ratios.

A more in-depth consideration of temperature effects on Si dissolution would be helpful to put their results into perspective as well. Since warming drives dissolution in the opposite direction from the OA effect examined here, the balance between climate-driven temperature increases and ocean pH reductions is critical to get right. They note this with the comment that "pH-related differences in opal dissolution are rather small compared to the influence of temperature", lines 546-547 in the SI. They also recognize that the effect sizes of OA and temperature more or less offset each other under the RCP 8.5 scenario (lines 558-560, SI). Clearly, Si dissolution is as sensitive (or possibly much more so) to warming as it is to OA changes. Since their mesocosms did not include temperature manipulations, getting this balance right cannot be based on measured experimental data, but only on the assumptions built into the model. Of course temperature will change the remineralization length scales of PON and POC as well, perhaps drastically, thus also altering the Si:N export ratios discussed here (and the Si:C export ratios that are not yet considered in this submission). A more thoughtful and prominent consideration of the possible relative influences of OA and warming is needed in the main text, not just hidden away in the Supplementary Information.

Minor issues:

Lines 15-16- awkward sentence

Line 36- silicoflagellates also need Si, change to 'in contrast to most other phytoplankton taxa'.

Lines 130-133- awkward sentence

Line 229- 'delusive' is awkward- change to 'incomplete'?

Responses to the referees' comments

Referee #1 (Remarks to the Author):

*Review : «Global decline of diatoms through ocean acidification » by Taucher and co-authors
Taucher and co-authors use (1) five in situ mesocosm experiments, and (2) a compilation of sediment trap data to demonstrate that ocean acidification increases the elemental ratio of Si (silicon) to N (nitrogen) in sinking biogenic material. They attribute this change in Si/N ratio to the slower chemical dissolution of biogenic silica when seawater pH is lower. They then use an earth system model of intermediate complexity (UVIC ESM) to test the significance of this potential effect (a pH-induced change in silica dissolution) in future projections. While diatoms have been shown to be resistant to, and even benefit from, ocean acidification, their model results suggest that the increase in Si/N ratio in response to future ocean acidification would result in the trapping of silicon in the deep ocean, to a more silicon-depleted surface ocean, and thus trigger an overall decline in diatom abundance in the long-term future ocean.*

I am in principle supportive, both of the results the authors present in their manuscript and of the methods they used, combining mesocosm experiments, a compilation of sediment trap data, and some specific model simulations. The manuscript is well written and easy to follow. The figures are of good quality. Their main result on the decrease in diatom abundance with ocean acidification is clearly at odds with the past literature and illustrates well how long-term time scales of ocean circulation superimposed on geochemical changes can redistribute nutrients between the surface and deep oceans and lead to counter-intuitive responses (ie. a decline in diatom abundance, whereas direct acidification effects may indeed be beneficial to diatoms).

That being said, I have some major reservations that prevent me from recommending this article for publication in Nature, at least in its current state. Because of my personal expertise, I'm mostly commenting here on the modelling aspects of the manuscript.

(1) On the choice to focus the model output analysis on the long-term (2200 instead of the end of the 21st century).

Most of the model projections that have been analysed and published so far in terms of the response of phytoplankton distribution to anthropogenic climate change focus on the end of the century (typically 2090-2100 or 2080-2100). These model studies suggest that the shifts in communities from projected reduction of nutrient supplies do favour smaller phytoplankton species over larger types such as diatoms (e.g. Bopp et al., 2005, Marinov et al. 2010, Dutkiewicz et al., 2013, Dutkiewicz et al. 2015).

Here, the authors have chosen to specifically focus their analysis on the simulated response at the end of the 22nd century – thus preventing a clear comparison to the above-mentioned model studies. I understand their choice is somehow dictated by the fact that their simulated response is linked to the long time-scale of the ocean circulation. But a much clearer justification of this choice is needed. I would also suggest to detail the time-scale of the response they obtain.

a) Comparison with other models (decreasing nutrients & shifts in phytoplankton)

Generally, the results from our model simulations are consistent with the previous studies mentioned by the reviewer, i.e. reduced nutrient supply (due to thermal stratification) lead

to a decrease in diatoms and a concomitant increase in smaller phytoplankton with lower nutrient requirements. Details on the model and its behavior under climate change simulations can be found in Kvale et al. (2020), and we would like to illustrate the most relevant information in the following plots (Fig. R1.1), showing the response of diatoms (left) and small phytoplankton (right) to simulated climate change. The black lines show the “control” model (standard climate change simulation, i.e. no pH effects on opal dissolution) and the red line the “OA” model (pH effects on opal dissolution, based on our experimental findings). Consistent with other models (e.g. (Dutkiewicz et al., 2013; Laufkötter et al., 2015; Marinov et al., 2010), we find that reduced nutrient supply in a warming ocean leads to a decrease in diatoms and favors phytoplankton with lower nutrient requirements (represented as half-saturation concentrations), i.e. corresponding to properties of small phytoplankton. By 2100, diatom biomass in the global ocean decreases by 18% compared to preindustrial conditions (“control”, black line), which is similar to results from previous studies (e.g. Bopp et al., (2005) or Laufkötter et al., (2015)). We added an additional section describing the consistency of our results with other global models to the supplementary discussion (section 3, lines 648-680)

Note that the model results presented in our manuscript focus on the OA effect, showing the difference between the simulations pH-sensitive opal dissolution, and the control simulations (i.e. “standard climate change”). The effect of pH on opal dissolution (“OA”, red line) amplifies the decrease in diatoms driven by climate change, reaching -27% in 2100 (compared to 18% in the “standard climate change” simulation). In the year 2200, the decrease in diatoms in the OA model is almost twice as strong as in the control model. At the same time, “small phytoplankton” that are mostly present in lower latitudes and can cope better with low nutrient concentrations, increase by 25% (year 2100).

Fig. R1.1: response of diatoms and small phytoplankton to climate change

b) Time-scale of results

As already pointed out by the reviewer, the choice to focus on 2200 was that the simulated impacts of pH-driven changes on opal dissolution and $\text{Si}(\text{OH})_4$ become increasingly relevant on the timescale of centuries, as these signals get transported with the global overturning circulation. We added a sentence in the method section to clarify this choice (lines 392-402) and also included more information on the time-scale of simulated

responses to the supplementary discussion (section 3, lines 648-680). However, as can be seen in Fig. 3b in the manuscript, which shows the temporal development of the response of diatom biomass to OA (driven by slower opal dissolution and loss of silicic acid from the surface ocean), the decline in diatoms is already apparent in 2100, when simulated diatom biomass is reduced by 10% (RCP8.5) and 8% (RCP6.0). Thus, the reference year (2100 or 2200) only affects the magnitude of the effect; qualitatively, the results are identical. It should also be noted that RCP8.5 simulates increasing CO₂ until 2250 – therefore the effects keep increasing until the end of our simulations. Under RCP6.0, CO₂ concentrations reach a plateau in 2150, thus the pH decline and the corresponding effect on Si:N slows down slightly in the 2nd half of the 22nd century.

(2) On the choice to focus the model output analysis on the response to Ocean Acidification only. The impact of an acidification-induced reduction in biogenic silica dissolution is indeed well illustrated by showing only differences between 2 model simulations (with / without the impact of pH changes on dissolution). But this leaves many questions unanswered, which at least merit some quantitative analysis and some discussion in the main text of the manuscript. What is the relative effect of decreasing pH vs. increasing temperature on opal dissolution and the subsequent biogeochemical impacts? How does the silica trapping effect (due to Si/N changes) compare to the other effects that have been discussed in the literature (due to the direct effects of ocean acidification on calcifying plankton types, or to the nutrient-induced shift towards smaller phytoplakton types?). In that regard, the supplementary discussion, addressing some of these questions, is much too qualitative and would benefit from a more through analysis.

(a) relative effect of decreasing pH vs. increasing temperature on opal dissolution?

Fig. R1.2: Comparison of warming and OA effects on opal dissolution

The simulated effect of OA on opal dissolution occurs on top of “background” climate change effects, incl. effects of warming on opal dissolution. Fig. R1.2 shows globally averaged vertical profiles of temperature (left) and pH (middle) under preindustrial

conditions and for the year 2200 (values are from the RCP8.5 simulation). The right panel shows a 1-D representation of the model parameterization for opal dissolution, applied to the corresponding temperature and pH profiles.

Averaged over the upper 1000m, the specific opal dissolution rate (global average) under preindustrial conditions amounts to $0.0008 \text{ [m}^{-1}\text{]}$, i.e. 0.08 % of sinking opal are dissolved per meter, resulting in a characteristic “Martin curve” of opal flux attenuation with depth (black line).

The specific dissolution rate of opal in the model (i.e. also in the standard model configuration) is temperature-dependent and roughly doubles for an increase in temperature of 10°C (consistent with other global models). For an increase in water temperature of $\sim 3^\circ\text{C}$ in the upper ocean until the year 2200 (averaged over the upper 1000m), the specific opal dissolution rate increases about 1.3-fold (from 0.08% to 0.11%), thereby leading to enhanced opal remineralization and a correspondingly stronger decline with depth in a warmer ocean (red dashed line) compared to preindustrial conditions.

At the same time, pH decreases by around 0.6 units in the upper 1000m of the ocean. According to our findings, opal dissolution rates are $\sim 57\%$ lower per unit decrease in pH, (i.e. a decrease by a factor of ~ 2.5). This effect of pH works in the opposite direction as the temperature effect, i.e. slowing down opal dissolution and thus weakening the decline of opal with depth. When considering this pH effect in isolation in our 1-D example, i.e. not accounting for the effects of warming (i.e. keeping temperature at preindustrial conditions), this drop in pH would decrease specific opal dissolution (in the upper 1000m) from 0.08% to 0.045%, i.e. reducing it by almost half (decrease by a factor of 1.8) (dashed blue line).

When considering both warming and acidification at the same time, the pH effect is still strong enough to over-compensate the temperature effect, i.e. opal dissolution in a warmer and lower-pH ocean is slower than under preindustrial conditions (green line).

Averaged over the upper 1000m, the specific opal dissolution rate amounts to 0.059%, which is notably lower than under preindustrial (0.08%) and warming-only (0.11%) conditions.

Altogether, opal dissolution under warmer and acidified conditions is slower, thus leading to an enhanced efficiency of opal export to the deep ocean.

We added a section and an additional figure to the supplementary discussion (section 2, lines 596-646), describing the relative effects of decreasing pH vs. increasing temperature on opal dissolution. Since this is a very important aspect, we also added the most relevant information to the main text (lines 136-139, 191-195, and lines 162-166 in the methods).

(b) Comparison to previously discussed effects of climate change

Reduced nutrients due to enhanced thermal stratification:

In the standard climate change simulation, the average decrease in Si(OH)_4 in the euphotic zone reaches -17% in the year 2100 and -23% in the year 2200. The decrease in NO_3 is similar, reaching about -14% in the year 2100 (-16% in the year 2200), which agrees well with projections from other global models (IPCC, 2019). The corresponding long-term nutrient trapping in the deep ocean induced by climate warming has also been observed in other global models (Moore et al., 2018).

In the model including OA effects on opal dissolution, the loss of Si(OH)_4 from the surface ocean is strongly amplified, reaching -27% in year 2100 and -46% in year 2200 (compared to -17% and -23% in the “standard climate change” simulation). According to these

numbers, the OA-driven decrease in opal dissolution almost doubles the long-term silica trapping in the deep ocean.

Effects on different phytoplankton types:

Our results from climate change simulations are consistent with those of previous studies (Dutkiewicz et al., 2013; Laufkötter et al., 2015; Marinov et al., 2010), e.g. a decrease in diatoms (due to decreasing nutrient supply) and increase in low-nutrient-adapted (small) phytoplankton (see Fig. R1 in response to comment #1).

Similar to the effect on Si(OH)_4 , the pH effect on opal dissolution amplifies the decline in diatoms under climate change: In the standard climate change simulation, the global decrease in diatoms in the euphotic reaches -18% and -26% (year 2100 and 2200, respectively), whereas it reaches -27% and -46% in the OA-simulation.

According to these numbers, the OA effects on opal dissolution and Si trapping amplify the global diatom decline by about 1.5 and 1.8-fold (year 2100 and 2200, respectively).

We added a discussion on the impacts of the silica trapping effect and how this compares to other effects (which have been previously discussed in literature) in section 3 of the supplementary discussion (lines 648-680).

Furthermore, we added an extensive discussion on the general limitations of the global model in section 4 of the supplementary discussion (lines 682-803).

(3) On the incorporation of the pH effect on biogenic silica dissolution

Finally, I have some reservations about how the effect of pH on dissolution, which is a key component of this study, is incorporated! As I understand it, biogenic silica export and associated dissolution in the water column is represented implicitly - with "vertical profiles of dissolved silica in the water column are instantaneously calculated based on biogenic silica production in the surface ocean." And silica dissolution is scaled "with pH changes from pre-industrial conditions for each model grid cell" in the new modified version. Are pH changes in the water column accounted for? Or only the pH changes at the ocean surface? This would be problematic and lead to an overestimation of the effect studied, as the pH changes at depth are likely to be much smaller than those simulated in the surface ocean. Simply put, I found the model description to be far too light and not consistent enough. The authors need to do a better job of detailing the new parameterization.

We thank the reviewer for pointing out that the description of the model, and how the parameterization of pH effects on opal dissolution were implemented, was not comprehensive enough.

The simulated effect of pH on opal dissolution accounts for vertical variability throughout the water column, i.e. it is scaled to the pH change at the respective depth (i.e. for each model box in the 3-dimensional grid). As mentioned by the reviewer, this is a critical point to consider, as the decrease in pH changes becomes smaller at greater depths (see Fig. R1.2).

We enhanced the model description in the manuscript to clarify this, particularly by describing in more detail how the parameterization for Si export and dissolution is scaled to future changes in pH (lines 368-387 in the methods).

(4) Literature suggested by the reviewer

Bopp, L. et al. (2005). Response of diatoms distribution to global warming and potential implications: A global model study. Geophysical Research Letters 32.

Dutkiewicz, S., Scott, J.R., and Follows, M.J. (2013). Winners and losers: Ecological and biogeochemical changes in a warming ocean. Global Biogeochemical Cycles 27, 463–477.

Dutkiewicz, S., Morris, J.J., Follows, M.J., Scott, J., Levitan, O., Dyhrman, S.T., and Berman-Frank, I. (2015). Impact of ocean acidification on the structure of future phytoplankton communities. Nature Climate Change 5, 1002–1006.

Marinov, I., Doney, S.C., and Lima, I.D. (2010). Response of ocean phytoplankton community structure to climate change over the 21st century: partitioning the effects of nutrients, temperature and light. Biogeosciences 7, 3941–3959.

We compared our modeling results with those from the references provided by the reviewer (see responses to comment #2).

References for response to reviewer #1

Bopp, L., Aumont, O., Cadule, P., Alvain, S., Gehlen, M. (2005) Response of diatoms distribution to global warming and potential implications: A global model study. *Geophysical Research Letters* 32.

Dutkiewicz, S., Scott, J.R., Follows, M.J. (2013) Winners and losers: Ecological and biogeochemical changes in a warming ocean. *Global Biogeochemical Cycles* 27, 463-477.

IPCC, (2019) IPCC Special Report on the Ocean and Cryosphere in a Changing Climate, in: Pörtner, H.-O., Roberts, D.C., Masson-Delmotte, V., Zhai, P., Tignor, M., Poloczanska, E., Mintenbeck, K., Alegría, A., Nicolai, M., Okem, A., Petzold, J., Rama, B., Weyer, N.M. (Eds.). Cambridge University Press, Cambridge, United Kingdom and New York, NY, USA.

Kvale, K., Keller, D.P., Koeve, W., Meissner, K.J., Somes, C., Yao, W., Oschlies, A. (2020) Explicit silicate cycling in the Kiel Marine Biogeochemistry Model, version 3 (KMBM3) embedded in the UVic ESCM version 2.9. *Geosci. Model Dev. Discuss.* 2020, 1-46.

Laufkötter, C., Vogt, M., Gruber, N., Aita-Noguchi, M., Aumont, O., Bopp, L., Buitenhuis, E., Doney, S.C., Dunne, J., Hashioka, T., Hauck, J., Hirata, T., John, J., Le Quéré, C., Lima, I.D., Nakano, H., Seferian, R., Totterdell, I., Vichi, M., Völker, C. (2015) Drivers and uncertainties of future global marine primary production in marine ecosystem models. *Biogeosciences* 12, 6955-6984.

Marinov, I., Doney, S.C., Lima, I.D. (2010) Response of ocean phytoplankton community structure to climate change over the 21st century: partitioning the effects of nutrients, temperature and light. *Biogeosciences* 7, 3941-3959.

Moore, J.K., Fu, W., Primeau, F., Britten, G.L., Lindsay, K., Long, M., Doney, S.C., Mahowald, N., Hoffman, F., Randerson, J.T. (2018) Sustained climate warming drives declining marine biological productivity. *Science* 359, 1139.

Referee #2 (Remarks to the Author):

Riebesell and colleagues provide evidence for slower silica dissolution with reduced pH, which leads to higher Si:N ratios in particulate matter sinking to depth. Using a Global Climate Model, the authors forecast how this abiotic change in dissolution affects the availability of Si in the surface ocean - resulting in a decline in diatom productivity in the future ocean.

This research is novel and challenges the current interpretation of the primary literature surrounding the effect of ocean acidification on diatoms - which up to this point, generally concludes that ocean acidification will benefit diatoms through physiological or ecological mechanisms. The authors use appropriate analyses for the meta-analysis and global analysis of sediment trap data. I am less familiar with the methods used in GCMs and suggest including a reviewer that frequently uses these models in their research. I was a little unclear about how to compare the confidence intervals for the probability density kernels, given they all overlap for some portion of the distributions. Some clarifications could be helpful here.

In principle, the probability density plots can be interpreted analogously to histograms. However, while histograms do not contain confidence intervals (e.g. to assess whether they differ from each other with statistical significance), this can be achieved with our approach, thereby separating real distribution patterns from observational noise.

Therefore, confidence limits for our Si:N distributions were obtained by applying a bootstrapping approach, where raw data of Si:N ratios were resampled (Efron and Tibshirani, 1993; Schartau et al., 2010). We first computed an ensemble of kernel density estimates based on resampled datasets. These were obtained by randomly subsampling each original data set with replacement (bootstrapping), thereby producing 1000 resampled datasets. For each of these, we computed kernel densities, thereby producing an ensemble of 1000 density distributions. Based on this, we computed standard deviations and 95% confidence intervals. As pointed out by the reviewer, these confidence intervals may overlap for some portion of the distributions. If, for a given range on the x-axis (i.e. range of Si:N ratios) the confidence intervals do not overlap, then a statistically significant difference can be assumed.

Fig R2.1: Example for interpretation of probability densities and their confidence intervals.

Fig. R2.1 shows the data from the Svalbard 2010 study. Si:N ratios between 0.1 and 0.2 (dotted box) were significantly higher under OA (red) compared to ambient CO₂ (blue), whereas lower Si:N ratios (approx. between 0.05 and 0.1; dashed box) were significantly lower under OA (red) compared to ambient CO₂ (blue). Altogether, this suggests a (statistically significant) shift from lower towards higher Si:N ratios under OA.

We added an additional sentence to the methods section (lines 311-314) and hope this helps readers in better interpreting the probability density plots.

While the results are novel and note-worthy, several of the points brought up in the current supplementary discussion deserve more discussion in the primary text. Most interesting to me is the limitations of the model used - such as how changes in the biology of the ecosystem could enhance or reduce the effects detected here. I would like to see how this mechanism plays out in other GCMs, given the results and conclusions are likely to be heavily influenced by the specifics of the model used. Model comparison would strengthen the findings of the forecasts.

We agree with the reviewer that it is important to consider how the biology of the ecosystem (and particularly differences among models) might enhance or reduce the effects detected here. However, a multi-model comparison is beyond the scope of our study. The core of our paper is the analysis of different observational datasets, which allowed us to reveal a mechanism that could be relevant for global diatom production and the oceanic Si cycle, particularly under ongoing ocean acidification. The goal of implementing this observed effect into an earth system model was to assess whether this effect could be of global relevance – and clearly, as pointed out by the reviewer, it should be emphasized that our model results only provide a first estimate and we hope that our study will be an incentive for other modeling groups to assess OA effects on opal dissolution with other global models, thereby assessing the potential variability of this effect among models.

An additional constraint for a comparison with more complex models (and the reason why we chose the UVic ESCM, an earth system model of intermediate complexity) is that the more complex the biology gets, the longer it takes to achieve an equilibrated deep ocean (model spinups on the order of 10,000 years). Therefore, models with more complex biology often address the upper ocean only and neglect the deep ocean (e.g. see studies by Dutkiewicz et al. (2021; 2015)). Since nutrient-trapping in the deep ocean is one of the key mechanisms in our OA simulations, we chose to apply a model that can be parameterized and spun-up relatively fast, in order to also achieve good agreements of the model results with observational data for the deep ocean.

Generally, it is reasonable to assume that the OA impacts on Si:N_{export} and opal dissolution would yield similar results if incorporated into other global models. The model we used (UVic ESCM) is similar to other models in terms of the structure of its ecosystem component, its skill in reproducing present-day conditions of biogeochemical tracers, and its behavior in climate change simulations, e.g. regarding future changes in nutrient supply (due to enhanced stratification) and corresponding decrease in phytoplankton biomass (cf. UVic projections by Keller et al. (2012) and Kvale et al. (2020)). Our finding of slower

opal dissolution under OA alters the attenuation of the downward opal flux (i.e. the vertical profiles of particulate opal and regenerated $\text{Si}(\text{OH})_4$). Since this is a chemical effect, this should not depend too much on the model specifics in terms of ecosystem structure – instead, the most important factor is likely how a global model simulates the spatial distribution of diatoms and opal production. This differs among various commonly used earth system models. As figure S2 in the supplementary material shows, the UVic model reproduces the global spatial distribution of surface $\text{Si}(\text{OH})_4$ quite well, thus giving us confidence that the spatial patterns of diatoms and opal production are also captured well by the model.

In theory, the OA-amplified decline of diatoms (as found in our model simulations) could trigger additional ecological changes (“knock-on effects” on plankton community structure), which might in turn modify carbon cycling and export fluxes. However, the low degree of ecological detail in most earth system models (including ours) does not allow to assess such complex knock-on effects. Therefore, it can only be speculated, based on theoretical knowledge, how such knock-on effects might look like. For instance, it is often suggested that a loss of diatoms might trigger changes in food-web structure, e.g. a loss of large copepods and/or shifts towards a longer food-chain (microzooplankton and microbial loop). Such a scenario might rather favor recycling in the surface ocean than export of larger fast-sinking particles, thereby possibly reducing the efficiency of the biological carbon pump. However, since common earth system models do not account for such ecological responses, such considerations are very speculative, which is why we refrained from discussing such implications of our findings on carbon export and potential climate feedbacks in too much detail.

We added a sentence to the manuscript to emphasize that our model results are robust and should be reproducible by other models (lines 204-205). Furthermore, we added an extensive discussion on the comparability of our model with other global models to the supplementary material, including model limitations related to biological processes not captured by the model (supplementary discussion sections 3 and 4, lines 648-803).

Overall, this is an extremely important and interesting study that highlights the complexity of global effects of ocean acidification and climate change on marine ecosystems. While there are some limitations in how much should be concluded from a single GCM, I think that the data are compelling enough to warrant publication and discussion.

We thank the reviewer for the kind words and helpful comments.

References for response to reviewer #2

Dutkiewicz, S., Boyd, P.W., Riebesell, U. (2021) Exploring biogeochemical and ecological redundancy in phytoplankton communities in the global ocean. *Global Change Biology* 27, 1196-1213.

Dutkiewicz, S., Morris, J.J., Follows, M.J., Scott, J., Levitan, O., Dyhrman, S.T., Berman-Frank, I. (2015) Impact of ocean acidification on the structure of future phytoplankton communities. *Nature Climate Change*.

Efron, B., Tibshirani, R.J. (1993) An introduction to the bootstrap.

Keller, D.P., Oschlies, A., Eby, M. (2012) A new marine ecosystem model for the University of Victoria Earth System Climate Model. *Geosci. Model Dev.* 5, 1195-1220.

Kvale, K., Keller, D.P., Koeve, W., Meissner, K.J., Somes, C., Yao, W., Oschlies, A. (2020) Explicit silicate cycling in the Kiel Marine Biogeochemistry Model, version 3 (KMBM3) embedded in the UVic ESCM version 2.9. *Geosci. Model Dev. Discuss.* 2020, 1-46.

Schartau, M., Landry, M.R., Armstrong, R.A. (2010) Density estimation of plankton size spectra: a reanalysis of IronEx II data. *Journal of Plankton Research* 32, 1167-1184.

Referee #3 (Remarks to the Author):

This manuscript presents a meta-analysis of five mesocosm experiments that examined the effects of ocean acidification on Si:N export ratios in the ocean. The authors are world experts in doing these types of large volume incubations, and the experiments and data analyses seem carefully and correctly done. Their major finding that exported Si:N decreases by 17% under OA conditions in four out of five of their experiments is surprisingly consistent, especially considering that their statistical methodology required them to average consolidated results from mesocosms with fairly widely varying CO₂ concentrations (lines 274-277). They make a good case that this OA effect could be a feedback mechanism not yet considered by biogeochemists. However, they also could do a much better job of recognizing the importance of other equally or more impactful environmental feedbacks on Si:N export, such as those arising from iron limitation and warming (see my comments below). I have four major comments, as well as some minor corrections and suggestions.

First of all, we thank the reviewer for her/his comprehensive and thoughtful comments, which we will address point-by-point in the following. Based on the comments, we also modified and complemented the manuscript and largely expanded the supplementary discussion to help the readership in interpreting our results. Altogether, this helped us to present our findings in a more robust way and thereby further improve our study.

My first major comment is that readers need to have access to the concentrations of particulate Si and N in the mesocosms and trap samples, not just the ratios. Obviously, a change in a ratio can be due to changes in either the numerator or the denominator, or to both. There is no way to tell whether their explanation that Si is dissolving more slowly at lower pH values is correct- it could just as easily be that N is remineralizing faster, or both could be happening at once. Please make available the raw data from which the ratios are calculated, and discuss the changes in the absolute fluxes of Si and N from the suspended plankton to the trap collections in both the mesocosms and (where possible) the meta-analyses of in situ sediment traps- not just the ratios.

We agree that it would indeed be additional powerful evidence if the pH effect on opal dissolution was detectable in absolute Si fluxes and not just the Si:N ratio. Thereby, the possibility of pH effects on N remineralization could be excluded as a possible driver of the observed Si:N responses.

However, absolute fluxes (i.e. a measure of the quantity of sinking material) in the ocean are highly variable with time and depth, whereas elemental ratios such as Si:N (measure of the quality of sinking material) are basically independent of the absolute mass fluxes and therefore much less variable in time and space. Accordingly, it is generally much more difficult to assess the influence of environmental drivers (here pH) on absolute fluxes than on elemental ratios. For the same reason, previous studies on global variability in opal export have used Si:C (and its vertical profile) as a proxy for differential dissolution/remineralization (Ragueneau et al., 2002; Ragueneau et al., 2006).

The same constraints apply to mesocosm experiments, for which the interpretation of absolute fluxes is very complex. Although some issues of field measurements are excluded (such as the influence of lateral advection), other processes affecting the mesocosm export flux do not allow for straight-forward interpretation of this data.

We will explain these issues in detail in the following paragraphs, both for mesocosm as well as for sediment trap data from the open ocean. Afterwards we will outline, why we are convinced that the different lines of evidence presented in our study provide a strong argument for pH effects on opal dissolution as the main driver of our findings.

Si:N_{export} and absolute fluxes of Si and N in the mesocosm studies

We will begin with showing the raw data of Si and N export from the different mesocosm experiments (Fig. R1) for which a significant effect of OA on Si:N_{export} was detected. Each data point represents the temporal average of a mesocosm (with a given OA scenario) during the study period.

Si vs. N as the driver of Si:N_{export}

Fig. R3.1: OA effects on Si:N_{export} (top row), absolute Si flux (middle row), and absolute N flux (bottom row) in the different mesocosm experiments (temporal average over the analyzed study period).

As can be seen in Fig. R3.1, absolute Si flux displayed a significant response to OA only in one study (Svalbard 2010). In 2 studies (Norway and Gran Canaria) there were no significant responses of either Si or N individually, but the Si:N response shows a clear effect. This illustrates, that Si:N ratios (and their response to OA) can display a very different pattern than absolute fluxes of Si and N.

The Sweden study provides an example for the confounding factors for interpreting absolute export fluxes: Here, N flux was lower under OA with Si flux not displaying a response – i.e. the Si:N response was actually driven by the denominator and not the numerator. At first glance, this could be interpreted as evidence that the OA effect on Si:N was driven by faster N remineralization (as suggested by the reviewer). However, this interpretation is misleading due to the particularities of mesocosm facilities and the

elemental fluxes and mass balances within these experimental enclosures: In the Sweden experiment, we have shown in earlier studies that OA resulted in higher biomass of mesozooplankton (copepods) and fish larvae (Boxhammer et al., 2018; Sswat et al., 2018). As a consequence, more N (but not Si) was stored in these elemental pools in the water column, thereby retaining N in the water column and leading to the lower export of N. This N is neither captured by export flux, nor by sampling of suspended POM (which excludes larger organisms such as mesozooplankton or fish larvae). Accordingly, the reduced export of N is not due to enhanced bacterial remineralization, but due to reallocation of N within the food web (in the water column). In contrast, the magnitude of Si export was similar across OA treatments. Thus, a lower amount of sinking biomass carried the same amount of opal. Our interpretation of these results with respect to the observed higher Si:N_{export} under OA is that more Si was retained per sinking particle due to slower dissolution of opal. The observed “no response” in Si is actually supporting this argument because in the absence of a pH effect on opal dissolution, one would expect reduced Si export, given that there were less resources (N) to support absolute export flux.

However, the influence of consumers on biogeochemical fluxes and pools in this study makes it impossible to prove this. This example illustrates how absolute export flux (in the mesocosms) can be influenced by a multitude of processes that control (a) how much of the suspended biomass is going to the sinking fraction and not only (b) how much of the sinking biomass is remineralized.

For the same reasons, we unfortunately cannot assess OA effects on flux attenuation based on “*changes in the absolute fluxes of Si and N from the suspended plankton to the trap*” as suggested by the reviewer. The data we have available are export fluxes (a rate measurement) and suspended matter concentrations in the water column (standing stock). Based on this data, we are not able to examine the vertical change in flux, as we have only measured flux at one depth (at the bottom of the mesocosm).

Ideally, one would need depth-resolved data of Si flux (and N) without any influence of advection etc. over a long enough time period at different seawater pH. With such data it would be possible to assess Si flux attenuation (due to chemical dissolution) and how this is controlled by pH, e.g. a stronger attenuation (larger “Martin’s b”) at higher pH. However, as shown in the following paragraph on the analysis of open ocean sediment traps, this is also difficult.

Note that in Fig. 1c in the manuscript, we show a visual comparison of export fluxes vs. suspended matter for Si:N ratios. In contrast to absolute fluxes, such a comparison is more meaningful for measures of particle quality such as Si:N, and was shown in the manuscript to demonstrate that the OA effect on Si:N did not arise during production of biomass.

Si:N_{export} and absolute fluxes of Si and N in open ocean sediment trap data

In order to assess whether pH effects on Si:N are unambiguously driven by Si dissolution, we also had a more in-depth look at the sediment trap data from a global compilation (Mouw et al., 2016). In contrast to mesocosm data, these datasets often span several depth levels, thus allowing to assess vertical changes in absolute fluxes and thereby – in theory – how much these are attenuated during sinking. However, field deployments are often affected by lateral advection of sinking biomass and temporal dynamics, making it difficult to unambiguously link particulate matter produced in the surface to particulate matter intercepted at depth. For instance, when sediment traps are deployed several weeks after

the bloom peak, then the biomass maximum has reached intermediate depths (the same can happen due to lateral advection of particulate matter). As a consequence, there is often no clear decrease of Si and/or N fluxes, i.e. lacking the typical "Martin's curve" pattern and thus not allowing to calculate a "Martin's b" coefficient to quantify the strength of flux attenuation (Fig. R3.2). In contrast, the Si:N ratio still maintains its characteristic vertical profile, as it only reflects the quality of sinking material (and how this changes with depth i.e. over time due to sinking) independent of the magnitude of sinking flux (which is subject to temporal variability in the surface ocean and thereby the vertical displacement of particle flux pulses). This is illustrated in Fig. R2 with examples from datasets in the North Atlantic (NABE) and North Pacific (OSP).

OSP: North-East Pacific, 50°N 145°W

NABE: North Atlantic, 48°N 21°W

Fig. R3.2: Examples of vertical profiles for Si:N and the corresponding Si and N fluxes from open ocean sediment traps in the North Atlantic (NABE) and North Pacific (OSP). Both stations are similar in temperature, but differ markedly in pH.

Note the strong relationships between depth and Si:N, even though there is large vertical variability in absolute Si and N fluxes. These non-steady vertical profiles of absolute fluxes (due to temporal variability or advection) do not permit to analyze how pH affects the flux attenuation of Si and N individually (i.e. to show if it is the numerator or denominator that is influenced by pH). In contrast, the general increase in Si:N with depth reflects how differential rates of Si dissolution (slower) and N remineralization (faster) alter the quality of the material while sinking down in the water column. It is also important to note that the Si:N ratio and its change with depth are essentially independent of absolute fluxes of Si and N (i.e. the same Si:N ratio can occur at high or low mass fluxes), which are often much more variable over time and in the vertical dimension. Accordingly, it is possible to examine vertical profiles of Si:N (and in our case how this is influenced by pH), even when there's no typical "Martin curve" relationship between depth and absolute Si and N fluxes. Thus, our approach is consistent with previous studies, which used vertical profiles of Si:C as a proxy for differential dissolution/remineralization of opal and its regional variability in the ocean (Ragueneau et al., 2002; Ragueneau et al., 2006).

Conclusion: how likely is it that pH effects on Si dissolution are the driver?

Regarding the reviewer's comment that the OA effect on $\text{Si:N}_{\text{export}}$ could also be driven by the denominator, i.e. "*that N is remineralizing faster, or both could be happening at once*": Based on the data from mesocosm experiments and ocean sediment traps, it is not possible to prove with 100% certainty that the observed Si:N responses are driven by pH effects on Si dissolution (as opposed to N remineralization). However, we are confident that the observed pH sensitivity of Si:N is driven by pH effects on silica (dissolution) for several reasons that we will explain in the following. As discussed in the supplementary material, previous chemical studies provided strong evidence for the pH dependence of opal dissolution (see Fig. R3.3 below).

As evident from the studies shown in Fig. R3.3, the pH dependence of opal dissolution is purely chemical. In the ocean, dissolution sets in once the organic coating around diatom shells has been degraded by bacteria (i.e. soon after cell death and thus close to the sea surface). In fact, it is worthwhile to consider that the measurement of biogenic silica (BSi) is based on this pH dependence: silicic acid is leached from particulate matter by digesting samples at high pH. Accordingly, it can be considered highly likely that this pH dependence of opal dissolution is valid for the entire ocean. In this regard, it is also notable that the magnitude of the pH effect inferred by our analysis (mesocosm experiments & ocean sediment traps) is strikingly consistent with the pH dependence from the chemical dissolution studies.

Figure from Greenwood et al., 2005, showing the pH dependence of the dissolution rate of biogenic silica produced by the diatom *Cylotella cryptica*. We marked the pH range relevant for present day (~8.2-7.7) and OA conditions (~7.5-7.8) with a red square. The authors find that the “log dissolution rate was found to scale linearly with pH with a slope of 0.38”. This corresponds to a decrease in dissolution rates by 60% (i.e. a factor of ~2.4) per unit pH, which is very close to the findings from our experiments (17% decrease in dissolution for a pH decrease of ~0.3 based on observed changes in $\text{Si:N}_{\text{export}}$; this would scale to a decrease of 57% per unit pH)

Figure from Van Capellen and Qiu, 1997, showing the effect of pH on silica dissolution rates at different degrees of undersaturation of the solution (x-axis). Note that seawater is generally very under saturated (0.8 to close to 1). The authors conclude that “the results clearly demonstrate the catalyzing effect of increasing pH (range 6-9) over the entire range of undersaturation. “

Figures from Dove and Elston, 1992. Left: Dissolution rates of quartz, which is chemically identical to biogenic silica (SiO_2), as a function of pH. Right: Predicted dissolution rates using a multiple regression approach for the various datasets analyzed in this study (left), additionally illustrating the role of salinity (i.e. higher dissolution rates with increasing salinity)

Fig. R3.3: Compilation of figures from studies assessing the pH dependence of opal dissolution, which we refer to in our study (Dove and Elston, 1992; Greenwood et al., 2005; Van Cappellen and Qiu, 1997)

Regarding possible effects of OA on N remineralization, the current consensus is that bacterial communities and organic matter remineralization are mostly resilient to ocean acidification (e.g. review by Hutchins and Fu (2017) and references therein). Results from studies that reported effects are quite variable and in most cases it was not possible to separate direct pH effects (e.g. pH effects on bacterial activity) from indirect effects mediated through pH-driven changes in quality and/or quantity of the organic matter substrate (Hutchins and Fu, 2017). Thus, there are currently no indications that OA will enhance N consumption of sinking organic matter.

Altogether, the various independent lines of evidence (mesocosms, ocean sediment traps, theoretical and empirical chemical studies) make the Si dissolution argument just much more likely than the other case. In the end, the interpretation of observational data is always dealing with likelihoods when studying natural systems. Thus, while we acknowledge that alternative explanations such as OA effects on N remineralization cannot be fully excluded, the consistency of available empirical datasets (as well as theoretical dissolution kinetics) suggests that the observed Si:N response is most likely attributable to Si dissolution.

Following the considerations above, we added more text to the manuscript to discuss possible alternative explanations for observed effects on Si:N and support our argumentation for the pH dependence of opal dissolution as the most likely explanation (lines 104-118 in the main text, lines 320-323 in the methods section, and an additional section in supplementary discussion, lines 562-594). We hope our work stimulates future work on this topic, e.g. targeted incubation experiments with sediment trap material and remineralization rates of different elements under different pH (and other potentially relevant environmental drivers).

A similar issue is the need to look for additional support for this Si dissolution trend using a different normalizer. Please plot the Si:C ratios as well as the Si:N ratios, and their changes from the suspended particulate matter to the trap collections (and the raw POC data they are based on, as above). These POC data should be easily available, since they did standard CN analyses to obtain the PON values. Considering Si:C ratios also would allow the authors to link Si fluxes directly to carbon export, something their ES model is not able to do (see lines 600-604 in the SI). A discussion of coupling to carbon export, or lack thereof, would greatly expand the biogeochemical and climate relevance of their take home message, which now focuses exclusively on Si export.

We agree with the reviewer that it would be interesting to also assess the relevance of our findings for the carbon cycle. However, the interpretation of Si:C responses to OA from our mesocosm data is difficult, as OA significantly affected C:N ratios in these studies (see Taucher et al. (2020)). This (biotic) effect is consistent with previous studies that reported sensitivity of many phytoplankton species to increasing CO₂, including corresponding changes in C:N of produced organic matter (e.g. due to enhanced carbon overconsumption). However, this effect complicates the interpretation of Si:C ratios. For instance, in the Sweden and Gran Canaria studies, OA increased C:N ratios. As a consequence, the Si:C effect is weaker (or absent) compared to Si:N (Fig. S3.4).

Si:C_{export}

Fig. R3.4: OA effects on $\text{Si:C}_{\text{export}}$ in the different mesocosm experiments

Furthermore, the main reason why we focused on the analysis of Si:N ratios is because of their relevance for global nutrient distributions (Brzezinski et al., 2002; Sarmiento and Gruber, 2006).

In the end, the mechanism that we revealed (slower opal dissolution under OA) should be independent of the C and N content of the sinking particles. It should also be noted that in the model simulations, the Si:N (or Si:C) ratios of export from the surface ocean are not directly affected by OA. The model parameterization scales opal dissolution with OA, which in turn affects Si:N (and Si:C) ratios and their increase with depth. However, the absolute fluxes of C and N (and their attenuation with depth) are not directly affected by OA in the model.

In terms of linking our findings to the carbon cycle, it is theoretically possible that the OA-driven decrease in opal dissolution indirectly affects the remineralization of C and N. It has been previously reported that opal mineral ballast can protect the organic material from remineralization (Moriceau et al., 2009). Thus, decreasing pH and the associated slower loss of the protective opal shell during sinking might theoretically decrease the remineralization rates of sinking C, N and P. However, this effect is not included in our model, as it would be very difficult to parameterize (not even the most sophisticated models account for this).

Another possible scenario by which the OA effect on opal dissolution could affect carbon export would be via the decline in diatoms as simulated by our earth system model. A loss of diatoms could (I) change the characteristics of sinking particulate matter, i.e. a larger proportion of non-diatom material may have lower sinking speeds (less mineral ballast) or higher remineralization rates (less protection of the silica shell) compared to diatom-derived material, or (II) alter the food-web structure of plankton communities with lower proportion of diatoms, e.g. changes in zooplankton composition and biomass, which would in turn affect carbon export.

Both mechanisms are highly complex and not included in our model simulations. Regarding (I): Due to controversial evidence on mineral ballasting, potential effects of opal on sinking velocities (due to the higher density of silica) are not included in most models, even the ecologically most sophisticated ones (Aumont et al., 2015; Dutkiewicz et al., 2021). Regarding (II): It is reasonable to assume that such a substantial decline of diatoms as found in our model simulations will trigger associated changes in food-web structure and composition of sinking organic matter, which will both have implications for carbon cycling and export. However, the low degree of ecological complexity in most earth system models (including ours) does not allow to make reliable predictions about such impacts.

Generally, it is often suggested that a loss of diatoms may reduce the efficiency of the biological carbon pump (slower sinking speed and faster remineralization of sinking organic matter of non-diatom origin) and may also trigger food-web changes, e.g. fewer large copepods and/or shifts towards a longer food-chain (microzooplankton and microbial loop), which would rather favor recycling in the surface ocean than export of larger fast-sinking particles. However, such considerations are very speculative and difficult to implement in a meaningful way in a global model given the underlying ecological complexity, which is why we refrained from discussing such implications of our findings on carbon export and potential climate feedbacks.

We added a discussion of the points raised above to the supplementary material (section 4 c and d, lines 765-803). Altogether, we hope that our study stimulates further research on this topic and to enable incorporating such ecological mechanisms into models, and thereby allowing us to make more reliable predictions on carbon cycling in future modeling studies.

The outcome of their model experiment suggests OA effects on Si dissolution in the Southern Ocean could have a major global effect on Si availability to diatoms through downstream reductions in upwelled Si at lower latitudes. This scenario has previously been raised a number of times in the literature for iron limitation, for instance in the Brzezinski et al. 2002 paper already cited in the text. This is because iron limitation greatly increases the Si:N and Si:C ratios of diatoms, by up to 200-300% in some cases (see older papers by Hutchins, Takeda and Brzezinski). Given that most models predict increases in iron supplies to the Southern Ocean in the future from a dustier climate and melting ice (see recent papers by Mahowald, for instance), isn't it possible that the relatively modest 17% reduction in sinking Si:N they model may be overwhelmed by much larger Si:N changes in diatoms due to iron limitation shifts? At the very least, a robust caveat is needed for the modeling component, as it apparently lacks dynamic terms for either changes in iron supply or their consequences for diatom Si:N ratios.

We agree with the reviewer that iron limitation (and its potential changes in the future) are an important factor to consider for the assessment of Si:N ratios. Biogenic opal production by diatoms in our model includes a parameterization for iron dependency of silicification, resulting in elevated Si:N ratios of production under iron limitation (see Kvale et al. (2020) for details). Accordingly, simulated $\text{Si:N}_{\text{export}}$ is elevated in large parts of the Southern Ocean (Fig. R3.5a). However, similar to most previous modeling studies, our model does not account for future changes in iron input. In fact, nutrient inputs from melting ice are not simulated by contemporary global models, and also dynamic dust deposition under climate change has only been recently implemented and tested in the latest generation of some global models (“CMIP6”, see e.g. Séférian et al., (2020). Generally, simulating the oceanic iron cycle (and reproducing observed spatiotemporal distributions) is still one the major current challenges in global biogeochemical modelling community (Séférian et al., 2020; Tagliabue et al., 2016).

Thus, the simulations with the UVic ESCM applied in our study does not simulate future changes in iron deposition, i.e. iron inputs to the ocean are fixed at preindustrial rates. However, the model includes a dynamic oceanic iron cycle and simulates an increase in iron availability in the surface ocean with climate change (Keller et al., 2012; Kvale et al., 2020). This increase in dissolved iron is a result of increasing stratification, which both reduces mixing and reduces export flux (and thereby also iron scavenging) and is also

found in other CMIP5 models (Fu et al., 2016). These simulated changes in dissolved iron allow us to estimate (to a certain degree) how predicted future changes in iron deposition in the Southern Ocean might alter silicification of diatoms, and thereby modify the OA-driven increase in $\text{Si:N}_{\text{export}}$.

As pointed out by the reviewer, both mechanisms (decrease in Si:N due to alleviated iron limitation and increase in Si:N due to slower opal dissolution under OA) would occur simultaneously under future climate change. Our model simulations account for both mechanisms, i.e. the pH-driven slower opal dissolution occurs on top of other “background effects” driven by climate change (such as lower Si:N due to more iron). In our simulations, dissolved iron in the Southern Ocean displays variable changes over the coming centuries (Fig. R3.5b). In regions where iron limitation is alleviated, $\text{Si:N}_{\text{export}}$ decreases due to lower silicification of diatoms (and vice versa; Fig. R3.5c), thereby counteracting to some extent the accelerated loss of Si from the surface ocean due to slower opal dissolution (at lower pH). However, in terms of effect size, the large-scale OA-induced increase in $\text{Si:N}_{\text{export}}$ is of similar magnitude (Fig. R3.5d)

Fig. R3.5: Model results in the Southern Ocean: (a) present-day $\text{Si:N}_{\text{exports}}$, (b) change in dissolved iron in the surface ocean until year 2200, (c) change in $\text{Si:N}_{\text{export}}$ until year 2200 in the original model, and (d) OA-driven change in $\text{Si:N}_{\text{export}}$ (year 2200, difference between model with OA-sensitive opal dissolution and original model).

It should also be noted that regional variabilities in Si:N in surface ocean due to factors controlling production (e.g. higher values due to iron limitation in the Southern Ocean due to higher Si:N of produced biogenic particles) are relatively small (max. values of ~ 3.0 in the Southern Ocean) compared to the vertical variability in the ocean as a result of differential dissolution/remineralization (reaching values of $>30-50$ in the deep ocean).

Thus, for the interpretation of our results, it is also important to differentiate between effects on Si:N during production and those occurring due to differential dissolution/remineralization of Si and N: Future changes in Si:N due to changes in iron would alter the total amount of exported Si, i.e. the magnitude of Si flux from the euphotic

zone (Fig R3.6). In contrast, the pH effect on Si dissolution alters the flux attenuation, i.e. the proportion of flux that is remineralized until the material reaches a certain depth. For the given example in Fig. R3.6, i.e. a 2-fold change in Si:N during production, the influence of both factors (changes in Si:N production and pH-driven change in opal dissolution) on deep ocean properties (Si:N and remaining opal) is on a similar order of magnitude, albeit with a slightly weaker effect of pH-dependent opal dissolution. If we consider this a scenario of alleviated iron limitation in the Southern Ocean (i.e. shift from dark to light blue) the OA-driven decrease in opal dissolution by 17% does not quite compensate for the 50% decrease in Si:N production (i.e. deep-ocean values of the dotted light blue do not reach values of the solid dark blue line). However, it should be emphasized that the relative size of the pH effect (i.e. the OA-driven decrease in the proportion of remineralized Si from sinking material) is independent of the magnitude of flux from the surface. Accordingly, the OA effect will lead to an enhanced loss of Si from the surface ocean, also when considering simultaneous changes in the Si:N ratios of particulate matter in the surface ocean. Thus, even if our model simulations had accounted for increased iron supply (e.g. via dust deposition) and correspondingly lower Si:N in the Southern Ocean, the influence of the OA-driven decrease in opal dissolution would occur on top of such background changes (similar to other “background climate change effects”, as described above). We added a section to the supplementary material to discuss future changes in iron limitation, and how these might potentially interact with the OA effect on opal dissolution (supplementary discussion, section 4b, lines 732-763).

Fig. R3.6: 1-D representation of the influence of changes in Si:N during production and OA effects on opal dissolution, using the parameterizations from the UVic ESCM for globally averaged vertical profiles of temperature and pH (values for the year 2100 under RCP8.5). Shown are scenarios for high Si:N (surface values of 2) and low Si:N (surface values of 1) of produced biogenic material that can occur e.g. due to differences in iron limitation (dark and light blue, respectively). Solid lines denote the original model and dotted lines the model with OA effects on opal dissolution.

A more in-depth consideration of temperature effects on Si dissolution would be helpful to put their results into perspective as well. Since warming drives dissolution in the opposite direction from the OA effect examined here, the balance between climate-driven temperature increases and ocean pH reductions is critical to get right. They note this with the comment that “pH-related differences in opal dissolution are rather small compared to the influence of temperature”, lines 546-547 in the SI. They also recognize that the effect sizes of OA and temperature more or less offset each

other under the RCP 8.5 scenario (lines 558-560, SI). Clearly, Si dissolution is as sensitive (or possibly much more so) to warming as it is to OA changes. Since their mesocosms did not include temperature manipulations, getting this balance right cannot be based on measured experimental data, but only on the assumptions built into the model. Of course temperature will change the remineralization length scales of PON and POC as well, perhaps drastically, thus also altering the Si:N export ratios discussed here (and the Si:C export ratios that are not yet considered in this submission). A more thoughtful and prominent consideration of the possible relative influences of OA and warming is needed in the main text, not just hidden away in the Supplementary Information.

We acknowledge that a more detailed explanation about the role of temperature on opal dissolution, and how this compares to the pH effect, is helpful for the readers. To illustrate this, we have visualized a 1-dimensional representation of temperature and pH effects on opal dissolution in our model (Fig. R3.7).

Fig. R3.7: Comparison of warming and OA effects on opal dissolution

Fig. R3.7 shows globally averaged vertical profiles of temperature (left) and pH (middle) under preindustrial conditions and for the year 2200 (values are from the RCP8.5 simulation). The right panel shows a 1-D representation of the model parameterization for opal dissolution, applied to the corresponding temperature and pH profiles. Averaged over the upper 1000m, the specific opal dissolution rate (global average) under preindustrial conditions amounts to $0.0008 \text{ [m}^{-1}\text{]}$, i.e. 0.08 % of sinking opal are dissolved per meter, resulting in a characteristic “Martin curve” of opal flux attenuation with depth (black line).

The specific dissolution rate of opal in the model (i.e. also in the standard model configuration) is temperature-dependent and roughly doubles for an increase in temperature of 10°C (consistent with other global models). For an increase in water temperature of $\sim 3^\circ\text{C}$ in the upper ocean until the year 2200 (averaged over the upper

1000m), the specific opal dissolution rate increases by about 1.3-fold (from 0.08% to 0.11%), thereby leading to enhanced opal remineralization and a correspondingly stronger decline with depth in a warmer ocean (red dashed line) compared to preindustrial conditions.

At the same time, pH decreases by around 0.6 units in the upper 1000m of the ocean. According to our findings, opal dissolution rates are ~57% lower per unit decrease in pH, (i.e. a decrease by a factor of ~2.5). This effect of pH works in the opposite direction as the temperature effect, i.e. slowing down opal dissolution and thus weakening the decline of opal with depth. When considering this pH effect in isolation in our 1-D example, i.e. not accounting for the effects of warming (i.e. keeping temperature at preindustrial conditions), this drop in pH would decrease specific opal dissolution (in the upper 1000m) from 0.08% to 0.045%, i.e. reducing it by almost half (decrease by a factor of 1.8) (dashed blue line). When considering both warming and acidification at the same time, the pH effect is still strong enough to over-compensate the temperature effect, i.e. opal dissolution in a warmer and lower-pH ocean is slower than under preindustrial conditions (green line).

Averaged over the upper 1000m, the specific opal dissolution rate amounts to 0.059%, which is notably lower than under preindustrial (0.08%) and warming-only (0.11%) conditions. Altogether, opal dissolution under warmer and acidified conditions is slower than in the preindustrial ocean, thus leading to an enhanced efficiency of opal export (and thereby loss of Si) to the deep ocean.

Since the considerations above are very important aspects for the interpretation of our results, we also added the most relevant information to the main text (lines 136-139, 191-195, and lines 162-166 in the methods).

We added an extensive discussion and an additional figure to the supplementary discussion (section 2, lines 596-646), describing the relative effects of decreasing pH vs. increasing temperature on opal dissolution.

Based on the numbers from the above example, we can better explain some of the statements in the manuscript (in the supplementary information), which have been raised by the reviewer:

“pH-related differences in opal dissolution are rather small compared to the influence of temperature”

This statement is valid in the context of present-day vertical gradients of temperature and pH, for which the temperature dependence has a roughly 10-fold larger effect on opal dissolution than the pH effect (as written in the SI: vertical gradients of 15-20°C affect the opal dissolution rate by a factor of 3-4 (i.e. 300-400%), whereas the vertical pH gradient of ~0.3 affects opal dissolution by ~20%, i.e. a factor of 1.2). Thus, temperature is the main driver for the vertical decrease in specific opal dissolution rates under present-day conditions (Fig. R3.7).

“the effect sizes of OA and temperature more or less offset each other under the RCP 8.5 scenario”

Unfortunately, our statement in the manuscript was a bit inaccurate. We wrote that “the effect size of both factors is of similar magnitude for the future changes of temperature and pH”. This was meant to clarify that, in contrast to present-day vertical gradients, the role

of temperature is not dominant anymore, i.e. that the role of pH becomes equally important. However, while the effect sizes are on the same order of magnitude, the effect of decreasing pH actually overcompensates for the effect of warming (Fig. R3.7). To avoid confusion in the manuscript, we change this part of the sentence to “the effect size of both factors is on a similar order of magnitude” (lines 614-615) and added a more extensive discussion on the roles of temperature and pH for opal dissolution to the supplementary material (section 2, lines 596-646).

“Since their mesocosms did not include temperature manipulations, getting this balance right cannot be based on measured experimental data, but only on the assumptions built into the model.”

It is correct that the considerations detailed above also depend on the model parameterization. The temperature sensitivity of opal dissolution follows the approach of other widely used models, whereas the incorporation of pH sensitivity into the model is based on our findings from the mesocosm OA experiments and previously published work on pH effects on opal dissolution (see response to the reviewer’s first comment). Clearly, some uncertainties remain for the exact effect size of both factors on opal dissolution, which could affect the balance between them. However, even if the (accelerating) effect of warming and the (slowing) effect of OA balance each other exactly out, this would still affect the interpretation of results from current earth system models, which all account for the effects of warming, but do not consider pH effects on opal dissolution. Thus, it can be expected that including this pH effect on opal dissolution in models will generally lead to a stronger loss of Si(OH)_4 from the surface ocean under simulated climate change.

“Of course temperature will change the remineralization length scales of PON and POC as well, perhaps drastically, thus also altering the Si:N export ratios discussed here (and the Si:C export ratios that are not yet considered in this submission).”

We agree that it is possible that temperature affects the remineralization length scales of C and N differently. Generally, opal dissolution rates in the ocean are much lower than those of organic material (C,N,P), which is the main reason why Si:N and Si:C ratios of particulate matter increase sharply with depth (Nelson et al., 1996; Ragueneau et al., 2002; Sarmiento and Gruber, 2006). This is also mirrored in vertical nutrient gradients of Si(OH)_4 and NO_3 in the ocean. A study based on laboratory experiments by Bidle et al. (2002) found that the C:Si remineralization ratio decreases with increasing temperature, i.e. the temperature effect on Si dissolution is larger than the temperature effect on C remineralization. This would mean that in warmer waters, the increase in Si:C with depth would be less pronounced than in colder waters. We are not aware of any experimental studies for Si:N or C:N remineralization ratios.

Our model (as other earth system models) applies the same temperature dependence for all elements and implementing a new model parameterization of temperature sensitivity for all elements was outside the scope of the present study. However, assuming that temperature (or also pH) affects the remineralization length scales of Si, C, and N at different degrees, this would theoretically alter the increase in Si:N and/or Si:C ratio with depth, and how ocean warming alters the vertical profile of opal flux attenuation (see figures above). Nevertheless the pH effect discussed in the present study comes on top of these warming-related changes and can be expected to generally amplify the loss of

Si(OH)₄ from the surface ocean under simulated climate change (which, however, might be additionally modified by temperature as correctly pointed out by the reviewer).

Minor issues:

Lines 15-16- awkward sentence

Line 36- silicoflagellates also need Si, change to 'in contrast to most other phytoplankton taxa'.

Lines 130-133- awkward sentence

Line 229- 'delusive' is awkward- change to 'incomplete'?

We corrected the minor issues identified by the reviewer.

In line 229, we changed the word to “deceptive”, as this emphasizes that our knowledge is based solely on small-scale experimental work and thus might not only be “incomplete” but in fact misleading. In the context of the present study, lab-based evidence resulted in a consensual view of diatoms as a winner of OA, whereas our large-scale perspective including earth system feedbacks (that do not matter in small-scale lab experiments) indicates the exact opposite.

References for response to reviewer #3

Aumont, O., Ethé, C., Tagliabue, A., Bopp, L., Gehlen, M. (2015) PISCES-v2: an ocean biogeochemical model for carbon and ecosystem studies. *Geosci. Model Dev.* 8, 2465-2513.

Bidle, K.D., Manganelli, M., Azam, F. (2002) Regulation of Oceanic Silicon and Carbon Preservation by Temperature Control on Bacteria. *Science* 298, 1980.

Boxhammer, T., Taucher, J., Bach, L.T., Achterberg, E.P., Algueró-Muñiz, M., Bellworthy, J., Czerny, J., Esposito, M., Haunost, M., Hellemann, D., Ludwig, A., Yong, J.C., Zark, M., Riebesell, U., Anderson, L.G. (2018) Enhanced transfer of organic matter to higher trophic levels caused by ocean acidification and its implications for export production: A mass balance approach. *Plos One* 13, e0197502.

Brzezinski, M.A., Pride, C.J., Franck, V.M., Sigman, D.M., Sarmiento, J.L., Matsumoto, K., Gruber, N., Rau, G.H., Coale, K.H. (2002) A switch from Si(OH)₄ to NO₃⁻ depletion in the glacial Southern Ocean. *Geophysical Research Letters* 29, 5-1-5-4.

Dove, P.M., Elston, S.F. (1992) Dissolution kinetics of quartz in sodium chloride solutions: Analysis of existing data and a rate model for 25°C. *Geochimica Et Cosmochimica Acta* 56, 4147-4156.

Dutkiewicz, S., Boyd, P.W., Riebesell, U. (2021) Exploring biogeochemical and ecological redundancy in phytoplankton communities in the global ocean. *Global Change Biology* 27, 1196-1213.

Fu, W., Randerson, J.T., Moore, J.K. (2016) Climate change impacts on net primary production (NPP) and export production (EP) regulated by increasing stratification and phytoplankton community structure in the CMIP5 models. *Biogeosciences* 13, 5151-5170.

Greenwood, J.E., Truesdale, V.W., Rendell, A.R. (2005) Toward an Understanding of Biogenic-silica Dissolution in Seawater – An Initial Rate Approach Applied between 40 and 90 °C. *Aquatic Geochemistry* 11, 1-20.

Hutchins, D.A., Fu, F. (2017) Microorganisms and ocean global change. *Nature Microbiology* 2, 17058.

Keller, D.P., Oschlies, A., Eby, M. (2012) A new marine ecosystem model for the University of Victoria Earth System Climate Model. *Geosci. Model Dev.* 5, 1195-1220.

Kvale, K., Keller, D.P., Koeve, W., Meissner, K.J., Somes, C., Yao, W., Oschlies, A. (2020) Explicit silicate cycling in the Kiel Marine Biogeochemistry Model, version 3 (KMBM3) embedded in the UVic ESCM version 2.9. *Geosci. Model Dev. Discuss.* 2020, 1-46.

Moriceau, B., Goutx, M., Guigue, C., Lee, C., Armstrong, R., Duflos, M., Tamburini, C., Charrière, B., Ragueneau, O. (2009) Si–C interactions during degradation of the diatom *Skeletonema marinoi*. *Deep Sea Research Part II: Topical Studies in Oceanography* 56, 1381-1395.

Mouw, C.B., Barnett, A., McKinley, G.A., Gloege, L., Pilcher, D. (2016) Global ocean particulate organic carbon flux merged with satellite parameters. *Earth Syst. Sci. Data* 8, 531-541.

Nelson, D.M., DeMaster, D.J., Dunbar, R.B., Smith Jr, W.O. (1996) Cycling of organic carbon and biogenic silica in the Southern Ocean: Estimates of water-column and sedimentary fluxes on the Ross Sea continental shelf. *Journal of Geophysical Research: Oceans* 101, 18519-18532.

Ragueneau, O., Dittert, N., Pondaven, P., Tréguer, P., Corrin, L. (2002) Si/C decoupling in the world ocean: is the Southern Ocean different? *Deep Sea Research Part II: Topical Studies in Oceanography* 49, 3127-3154.

Ragueneau, O., Schultes, S., Bidle, K., Claquin, P., Moriceau, B. (2006) Si and C interactions in the world ocean: Importance of ecological processes and implications for the role of diatoms in the biological pump. *Global Biogeochemical Cycles* 20.

Sarmiento, J.L., Gruber, N. (2006) *Ocean biogeochemical dynamics*. Princeton University Press.

Séférian, R., Berthet, S., Yool, A., Palmiéri, J., Bopp, L., Tagliabue, A., Kwiatkowski, L., Aumont, O., Christian, J., Dunne, J., Gehlen, M., Ilyina, T., John, J.G., Li, H., Long, M.C., Luo, J.Y., Nakano, H., Romanou, A., Schwinger, J., Stock, C., Santana-Falcón, Y., Takano, Y., Tjiputra, J., Tsujino, H., Watanabe, M., Wu, T., Wu, F., Yamamoto, A. (2020) Tracking Improvement in Simulated Marine Biogeochemistry Between CMIP5 and CMIP6. *Current Climate Change Reports* 6, 95-119.

Sswat, M., Stiasny, M.H., Taucher, J., Algueró-Muñiz, M., Bach, L.T., Jutfelt, F., Riebesell, U., Clemmesen, C. (2018) Food web changes under ocean acidification promote herring larvae survival. *Nature Ecology & Evolution*.

Tagliabue, A., Aumont, O., DeAth, R., Dunne, J.P., Dutkiewicz, S., Galbraith, E., Misumi, K., Moore, J.K., Ridgwell, A., Sherman, E., Stock, C., Vichi, M., Völker, C., Yool, A. (2016) How well do global ocean biogeochemistry models simulate dissolved iron distributions? *Global Biogeochemical Cycles* 30, 149-174.

Taucher, J., Boxhammer, T., Bach, L.T., Paul, A.J., Schartau, M., Stange, P., Riebesell, U. (2020) Changing carbon-to-nitrogen ratios of organic-matter export under ocean acidification. *Nature Climate Change*.

Van Cappellen, P., Qiu, L. (1997) Biogenic silica dissolution in sediments of the Southern Ocean. II. Kinetics. *Deep Sea Research Part II: Topical Studies in Oceanography* 44, 1129-1149.

Reviewer Reports on the First Revision:

Referees' comments:

Referee #1 (Remarks to the Author):

The authors have satisfactorily addressed the 3 main points raised in my review of the initial manuscript (i.e. (1) the focus on 2200 instead of 2100 as in most other studies, (2) the lack of discussion on the other effects than OA on diatoms abundance, and (3) the light description of the model used in the study).

The text has been changed accordingly – but most of the additional analyses and explanations are restricted to the supplementary material. I would however suggest including in the main text

- the projected estimates for year 2100 (and in addition to those for 2200), to facilitate subsequent model comparisons,

- the projected estimates, discussed now in the supplementary, of the warming-only effects (without considering OA) - for instance on Figure 3b.

Referee #3 (Remarks to the Author):

I appreciate the very in-depth responses to the review comments from the authors. I agree that their work represents a significant development in our understanding of how nutrient biogeochemistry will respond to ocean acidification. My remaining questions and comments are summarized as follows:

1) Thanks to the authors for pointing out in detail that the effects of ocean acidification on Si:N ratios are independent of the absolute fluxes of Si and N. Of course this doesn't mean that absolute fluxes, and Si:N production ratios in the surface ocean, are irrelevant to Si trapping- this is what primes the Si pump, so to speak. They have done a better job of recognizing this in the revised manuscript with some consideration of factors like changing iron availability and diatom dominance that will affect production ratios, and so will be superimposed on the Si solubility process they are modeling (or vice versa, as they prefer to consider it). I still think it is necessary to provide readers with access to the absolute Si and N values used to calculate the Si:N ratios (and Si:C ratios- see below) they focus on- without these underlying measured values, the unitless ratios leave open questions like the ones I posed about what processes could be driving the observed pH effects. Please provide these original, absolute data to readers for all of the incubation and sediment trap samplings in the form of a table in the supplementary materials. Giving readers access to the original data rather than only to derived ratios is a fundamental requirement for modern oceanographic publications- or should be.

2) As they correctly state in their response letter, they cannot evaluate vertical changes in Si:N fluxes, as their tank experiments only captured fluxes at a single depth. It is harder to accept that the reason particulate Si:N ratios couldn't be accurately measured near the surface is because a N mass balance couldn't be achieved due to zooplankton grazing. There are simple protocols available for doing quantitative net tows, which would have allowed quantification and summing of zooplankton-associated N along with the suspended particulate (phytoplankton) measurements. In fact,

zooplankton consume diatom Si along with N (although they don't assimilate it, of course), so a true mass balance for this nutrient central to their story in the upper layer of their tanks is not possible either.

3) The authors responded at length to my questions about linking the Si dissolution effect they are examining with the biological pump by considering Si:C ratios in addition to Si:N ratios. Their response is essentially that Si fluxes cannot be coupled with C fluxes because C:N ratios sometimes increase with acidification (although this only happened in 2 out of their 5 mesocosm experiments). This is a bit disingenuous- since as they say Si:N ratio changes are independent of absolute fluxes of Si and N, the same thing must be true of Si and C as well, so even if there is relatively more C being fixed by diatoms in the euphotic zone, differential remineralization of Si at low pH should still be evident in the Si:C flux ratio. At the very least, they need to add the Si:C data they showed me in their response letter into the supplementary material in the paper (and including the underlying BSi and POC data, see above), discuss it thoroughly, and add this as another caveat to the text. As it is, in their response to my comment they cite sections 4 c and d in the SM as their relevant revision, but actually this section is exclusively about uncertainties surrounding the effects of opal ballasting- a related but different issue. In addition to this very appropriate ballasting discussion, the authors also need to have the same conversation we are having here, and if pH effects on Si:C ratios are weaker (or absent) than those on Si:N ratios due to acidification effects (as they say in the response letter), this needs to be clearly stated and explained in the main manuscript as well. Otherwise, readers will definitely be left to wonder why the presentation and discussion ignores Si:C ratios and C export in this paper, which of course is at least as important to ocean biogeochemistry as nutrient export ratios.

Responses to the referees' comments

Referee #1 (Remarks to the Author):

The authors have satisfactorily addressed the 3 main points raised in my review of the initial manuscript (i.e. (1) the focus on 2200 instead of 2100 as in most other studies, (2) the lack of discussion on the other effects than OA on diatoms abundance, and (3) the light description of the model used in the study).

The text has been changed accordingly – but most of the additional analyses and explanations are restricted to the supplementary material. I would however suggest including in the main text

- the projected estimates for year 2100 (and in addition to those for 2200), to facilitate subsequent model comparisons,*
- the projected estimates, discussed now in the supplementary, of the warming-only effects (without considering OA) - for instance on Figure 3b.*

We are glad that we could satisfactorily address all issues raised by the reviewer, and we acknowledge that additional results from our model simulations will be very helpful for model comparisons in the future, e.g. to simulate OA effects on opal dissolution with different global models, and thereby assessing the potential variability of this effect among models.

However, it is unfortunately not so easy to accommodate this quite comprehensive information in the main text. The Nature formatting guidelines regarding text length and number of figures do not allow us to expand the main text in order to include a comprehensive presentation and results from both models (“warming only” effects in addition to results from the model with warming+OA effects on Si) for both RCP scenarios, i.e. 4 model simulations in total.

In this regard, it is important to note that in the current state, our presentation and visualization of model results in the main text shows the net effect of OA-sensitive opal dissolution (Δ_{OA}), which is quantified as the difference between (a) the model including OA effects on opal dissolution and (b) the standard model configuration (“warming only”). This is now also better explained in the Methods and the corresponding text passages. For this reason, replacing Fig. 3b (now showing Δ_{OA}) with a plot including “warming-only” effects (as suggested by the reviewer) would require to plot temporal changes (Δ_{time} , compared to preindustrial conditions) instead of Δ_{OA} (the effect size due to pH-dependent Si dissolution). Since the other panels in Fig. 3 also show Δ_{OA} , this would likely confuse readers. Analogously, numbers for the year 2100 in the main text would only be helpful for comparison with previous modeling studies if they also include the Δ_{time} of all 4 model simulations (as commonly reported), and not just Δ_{OA} (as currently, to put the focus on the newly discovered OA effect on Si dissolution). The same reasoning for limitations of text length apply here, unfortunately.

Nevertheless, we fully agree that this information is helpful, particularly for other modelers. Therefore, we now provide detailed results in the extended data (an additional figure and table; see below), which contain all the relevant information, i.e. model behaviour under climate change of the standard model (“warming only”) and the model

including OA effects on Si dissolution, both for temporal changes (Δ_{time}) as well as Δ_{OA} , as well as for numbers for different reference years (2100 and 2200). The additional figure and table presents the results more prominently compared to the previous version of the manuscript, so that they are not “hidden” in the supplementary discussion anymore.

In this regard, we emphasize that according to the Nature publishing guideline, the Extended Data would be integrated into the final version of the paper, in case of acceptance and publication. More specifically “Extended Data are included in the online version as well as the PDF” and “Extended Data are an integral part of the paper and only data that directly contribute to the main message should be presented.”

We think that the information required by the reviewer exactly matches these criteria and are convinced that this additional and easily accessible information will facilitate subsequent model comparisons.

	Year	Si(OH) ₄	NO ₃	Diatoms	Phyto _{small}
RCP6.0					
Change relative to preindustrial conditions (year 1750)					
Standard model (no OA impacts on Si)	2100	-13%	-8%	-14%	-2%
	2200	-19%	-10%	-18%	-1%
Including OA impacts on Si dissolution	2100	-21%	-8%	-22%	+5%
	2200	-33%	-10%	-29%	+9%
Net OA effect (including vs. excluding OA impacts on Si dissolution)					
Net OA effect (Δ_{OA})	2100	-9%	0%	-8%	+7%
	2200	-17%	0%	-13%	+10%
RCP8.5					
Change relative to preindustrial conditions (year 1750)					
Standard model (no OA impacts on Si)	2100	-15%	-10%	-18%	-2%
	2200	-22%	-13%	-25%	0%
Including OA impacts on Si dissolution	2100	-24%	-10%	-26%	+6%
	2200	-42%	-12%	-45%	+19%
Net OA effect (including vs. excluding OA impacts on Si dissolution)					
Net OA effect (Δ_{OA})	2100	-11%	0%	-10%	+8%
	2200	-27%	0%	-26%	+18%

Table for “Extended Data”: Comparison of global scale impacts of OA-driven slowdown of opal dissolution vs. other climate change impacts in our simulations with the UVic model. (Note that caption in the manuscript includes related discussion, e.g. comparison to previous work)

Figure for “Extended Data” : Future changes in (a) Si(OH)_4 and (b) diatom biomass in the surface ocean relative to preindustrial conditions. Shown are results from simulations with the standard model configuration, i.e. excluding OA-effects on silica dissolution (dashed lines), and the model including OA-effects on silica dissolution (solid lines) for RCP6.0 and RCP8.5 emission scenarios.

Referee #3 (Remarks to the Author):

I appreciate the very in-depth responses to the review comments from the authors. I agree that their work represents a significant development in our understanding of how nutrient biogeochemistry will respond to ocean acidification. My remaining questions and comments are summarized as follows:

1) Thanks to the authors for pointing out in detail that the effects of ocean acidification on Si:N ratios are independent of the absolute fluxes of Si and N. Of course this doesn't mean that absolute fluxes, and Si:N production ratios in the surface ocean, are irrelevant to Si trapping- this is what primes the Si pump, so to speak. They have done a better job of recognizing this in the revised manuscript with some consideration of factors like changing iron availability and diatom dominance that will affect production ratios, and so will be superimposed on the Si solubility process they are modeling (or vice versa, as they prefer to consider it). I still think it is necessary to provide readers with access to the absolute Si and N values used to calculate the Si:N ratios (and Si:C ratios- see below) they focus on- without these underlying measured values, the unitless ratios leave open questions like the ones I posed about what processes could be driving the observed pH effects. Please provide these original, absolute data to readers for all of the incubation and sediment trap samplings in the form of a table in the supplementary materials. Giving readers access to the original data rather than only to derived ratios is a fundamental requirement for modern oceanographic publications- or should be.

We agree that fluxes and Si:N ratios in the surface ocean are just as important as processes occurring deeper down in the water column (e.g. Si dissolution), and that both might be differentially affected by future climate change. We hope that the additional discussion (that emerged from this constructive review process), e.g. on iron effects in the Southern Ocean, helps to illustrate the complexity of these issues to the readership.

Following the reviewer's advice, we have now included the raw data of Si, N, C for water column and sediment traps, both for absolute fluxes and for the respective ratios, in Extended Data Table 2. The table includes temporal averages that were used in the presented analysis (e.g. effect sizes, linear regressions). The original full dataset (timeseries data for each mesocosm and experiment) contains more than 1000 measurements and is therefore much too large for a display item. Instead, it is provided as a datasheet in the Supplementary Data, as well as on the PANGAEA data repository (uploaded upon acceptance of the manuscript).

2) As they correctly state in their response letter, they cannot evaluate vertical changes in Si:N fluxes, as their tank experiments only captured fluxes at a single depth. It is harder to accept that the reason particulate Si:N ratios couldn't be accurately measured near the surface is because a N mass balance couldn't be achieved due to zooplankton grazing. There are simple protocols available for doing quantitative net tows, which would have allowed quantification and summing of zooplankton-associated N along with the suspended particulate (phytoplankton) measurements. In fact, zooplankton consume diatom Si along with N (although they don't assimilate it, of course), so a true mass balance for this nutrient central to their story in the upper layer of their tanks is not possible either.

We fully agree that accurate mass balances of Si and N (and other elements) would have facilitated data interpretation. However, we note that elemental mass balance estimates in marine ecosystems remain challenging. Even in enclosed mesocosm systems with discrete measurements of all relevant parameters, achieving a closed mass balance is more difficult than one would expect in theory. Major reasons are e.g. (i) the small-scale vertical variability of particulate matter and/or nutrient concentrations (e.g. due to thermal stratification), that may often not be resolved by the sampling methods, and (ii) error propagation due to accumulating uncertainties of several measurements, each with their own precision and accuracy (e.g. for nitrogen: NO_3 , NH_4 , PON, DON, and N in micro- and mesozooplankton).

This is also a reason, why we consider our data on vertical fluxes (and their elemental composition), which our findings are mainly based on, to be the most robust. In the particular case of our experiments, sediment trap measurements collected all the particulate matter from the mesocosm enclosure, corresponding to a volume of 30-50 m^3 (see Extended Data Fig. 1 for illustration). Thus, this measurement is not affected by vertical variability within the overlying water column (as mentioned above) and also has a higher measurement accuracy and precision: there was usually more than sufficient amount of particulate matter to achieve a signal-to-noise ratio for the measurement. In contrast, when water column concentrations were low (e.g. periods of low productivity), the signal-to-noise ratio of particulate matter measurements was much larger. Therefore, data on vertical fluxes (and their elemental composition) can be considered more robust than mass balance estimates within the water column.

Clearly, the importance of mesozooplankton on elemental cycling should not be underestimated, but at the same time, is challenging to quantify accurately based on the available data. We note that net tows were conducted quantitatively, using the same net (with a given diameter and mesh size) over the same depth interval in each experiment. However, data from net catches commonly show large variability (both in mesocosm as well as on ship expeditions), e.g. due to small-scale patchiness and escaping organism that affect catch efficiency. This makes it particularly challenging to account for mesozooplankton in mass balance estimates.

Furthermore, the common practice of processing net catches is to count organisms and then use conversion factors (from literature) to compute the equivalent of biomass (e.g. in units C or N). Clearly, this approach is not accurate enough when assessing subtle changes in mass balances, and assuming fixed conversion factors would also be inadequate when investigating elemental composition (as done in our study). In this case, it might have been more practicable to measure the biomass and elemental composition of bulk net samples. However, since the zooplankton work in the mesocosm experiments was focused on ecology and diversity, the net catches were used for microscopic analysis (counting) and therefore unfortunately not available for determination of biomass and elements.

Nevertheless, while we acknowledge the various issues described above, they mostly affect the interpretation of suspended matter in the water column and/or elemental mass balances (as the reviewer correctly points out). Since our main conclusions are based upon the measurement of vertical fluxes from sediment traps and their elemental composition – which as described above are least prone to measurement errors (and do not require closed

mass balance calculations for the overlying water column) – we are confident that our findings on $\text{Si:N}_{\text{export}}$ and how it is influenced by ocean acidification are robust and reproducible. Thus, we also hope that our work is an incentive for other researchers to carry out more targeted experiments, e.g. to assess the role of different environmental factors (pH, temperature, oxygen) on degradation and Si:C:N:P composition of sinking particulate matter in dedicated future work.

3) The authors responded at length to my questions about linking the Si dissolution effect they are examining with the biological pump by considering Si:C ratios in addition to Si:N ratios. Their response is essentially that Si fluxes cannot be coupled with C fluxes because C:N ratios sometimes increase with acidification (although this only happened in 2 out of their 5 mesocosm experiments). This is a bit disingenuous- since as they say Si:N ratio changes are independent of absolute fluxes of Si and N, the same thing must be true of Si and C as well, so even if there is relatively more C being fixed by diatoms in the euphotic zone, differential remineralization of Si at low pH should still be evident in the Si:C flux ratio. At the very least, they need to add the Si:C data they showed me in their response letter into the supplementary material in the paper (and including the underlying BSi and POC data, see above), discuss it thoroughly, and add this as another caveat to the text. As it is, in their response to my comment they cite sections 4 c and d in the SM as their relevant revision, but actually this section is exclusively about uncertainties surrounding the effects of opal ballasting- a related but different issue. In addition to this very appropriate ballasting discussion, the authors also need to have the same conversation we are having here, and if pH effects on Si:C ratios are weaker (or absent) than those on Si:N ratios due to acidification effects (as they say in the response letter), this needs to be clearly stated and explained in the main manuscript as well. Otherwise, readers will definitely be left to wonder why the presentation and discussion ignores Si:C ratios and C export in this paper, which of course is at least as important to ocean biogeochemistry as nutrient export ratios.

Our statement that Si:N ratio changes are independent of absolute mass fluxes (quantity) is still valid, also in the case of Si:C. However, in this context, the C:N ratio (i.e. another factor affecting the quality of sinking particles) complicates the interpretation. As the reviewer correctly summarizes, the coupling of Si fluxes to C fluxes (Si:C) is more difficult than for N fluxes (Si:N), because the C:N ratio was sensitive to ocean acidification (OA). In fact, this was the case in 4 out of 5 experiments (see figure below). Notably, however, the effect of OA on C:N was much more variable, thus also affecting Si:C to varying extents: In Svalbard, there was a positive, but relatively small effect on C:N, resulting in a slightly slower Si:C response than for Si:N. In Sweden and Gran Canaria, the positive OA effect on C:N was sufficiently large to decrease Si:C to a degree (compared to Si:N) that the effect size is not statistically significant anymore. In contrast, OA lowered C:N in the Norway experiment, thereby even slightly enhancing the Si:C response compared to Si:N. We created a plot that includes OA effects on Si:C and C:N (in addition to Si:N) and shows how variable C:N responses counteract or amplify effects on Si:C compared to Si:N, thus resulting in more variable and less clear OA effects on Si:C (Extended Data Fig. 2; also see below). The raw data is also included in Extended Data Table 2, which we added to the manuscript (see reply to comment 1).

Note that in contrast to the OA effect on Si (slower chemical dissolution), the effect on C:N is biotic and likely results from responses of primary producers and heterotrophic consumers, which together leave an imprint on the C:N sinking out of the productive surface layer. Therefore, it can be assumed that this OA effect on C:N of freshly produced material is rather restricted to the surface, whereas the OA effect on Si occurs throughout the entire water column. Nevertheless, the argumentation here would be similar to the previous discussion with the reviewer regarding iron effects on Si:N (now included in the manuscript), that is, the importance to differentiate between effects in the surface ocean (production) and those occurring throughout the water column (e.g. dissolution): If there are OA-effects on in C:N in the surface, this would clearly also affect Si:C. Accordingly, OA effects on Si dissolution in the water column would be superimposed on the initial Si:C of particulate matter produced at the surface, i.e. enhancing the preservation of Si compared to C of particles sinking through the water column (given as the increase in Si:C with depth). We included a statement on the different effects on Si:C in the main text (lines 125-127) and refer to the new figure to make the readers aware of the issue (the caption contains a brief discussion of the most important points above to address the points raised by the reviewer).

Extended Data Fig. 2: Comparison of OA impacts on different elemental ratios (Si:N, C:N, and Si:C) of sinking biogenic matter. Note that effects on Si:C display a larger variability than Si:N due to additional (but variable) responses of C:N ratios. In contrast to the OA effect on Si (slower chemical dissolution), the shifts in C:N are biotically driven, resulting from responses of primary producers and heterotrophic consumers, which together leave an imprint on the C:N sinking out of the productive surface layer. Depending on the direction and magnitude of the OA effect on C:N, this offsets or amplifies the effect on Si:C compared to Si:N. However, since OA effects on C:N are driven by biotic processes and linked to freshly produced material, it can be assumed that this effect is mostly restricted to the surface ocean, whereas the OA effect on Si occurs throughout the entire water column (see Extended Data Fig. 3). It can thus be expected that the OA-driven

decrease in opal dissolution will enhance the preservation of Si compared to C, thereby increase Si:C ratios of particles sinking through the water column.

While the more variable effects on Si:C compared to Si:N should not affect our findings in terms of the loss of Si(OH)_4 from the surface ocean (absolute, as well as relative to NO_3) and the resulting decline in diatoms (due to lower nutrient availability), these effects could potentially alter carbon export and the biological pump.

The major reason why we have refrained from additionally including this in our earth system model simulations is that OA effects on C:N differed strongly among experiments, and the underlying mechanism is biotic – and therefore likely much more complex and variable than the pH effect on chemical silica dissolution. Thus, the validity of such a model would be very uncertain at this stage. In our view, this would require much more information and a better understanding of the underlying mechanisms to conscientiously implement this mechanism into a global model (which would easily be a study in itself).

Reviewer Reports on the Second Revision:

Referees' comments:

Referee #1 (Remarks to the Author):

I thank the authors for following my recommendations. I fully understand that given the length restrictions imposed by the Letter format in Nature, it is not possible to include figures and discussions of the climate-only effect in the main text. The solution proposed by the authors, figure and table in the Extended Data, seems to me acceptable and will allow in the future an easier comparison with other modeling studies.

Referee #3 (Remarks to the Author):

The authors have done a good job of addressing my remaining reservations, and the paper is much improved.